# Validation of an ocean shelf model for the prediction of mixed-layer properties west of Sardinia (Mediterranean Sea)

Reiner Onken[1]

[1]Helmholtz-Zentrum Geesthacht Centre for Materials und Coastal Research (HZG), Max-Planck-Straße 1, 21502 Geesthacht, Germany

*Correspondence to:* Reiner.Onken@hzg.de

**Abstract.** The *Regional Ocean Modeling System* (ROMS) has been employed to explore the sensitivity of the forecast skill of mixed-layer properties to the initial conditions, boundary conditions, and vertical mixing parameterisations. The initial and lateral boundary conditions were provided by the *Mediterranean Forecasting System* (MFS) or by the MERCATOR global ocean circulation model via one-way nesting; the initial conditions were additionally updated by the assimilation of observations. Nowcasts and forecasts from the weather forecast models COSMO-ME and COSMO-IT, partly melded with observations, served as surface boundary conditions. The vertical mixing was parameterised by the GLS (*Generic Length Scale*) scheme (Umlauf and Burchard, 2003) in four different setups. All ROMS forecasts were validated against observations which were taken during the REP14-MED oceanographic survey to the west of Sardinia. Nesting ROMS in MERCATOR and updating the initial conditions by data assimilation provided the best agreement of the predicted mixed-layer temperature and the mixed-layer depth with time series from a moored thermistor chain. Further improvement was obtained by the usage of COSMO-ME atmospheric forcing which was melded with real observations, and by the application of the k-$\omega$ vertical mixing scheme with increased vertical eddy diffusivity. The predicted temporal variability of the mixed-layer temperature was reasonably well correlated with the observed variability while the modelled variability of the mixed-layer depth exhibited only agreement with the observations near the diurnal frequency peak. For the forecasted horizontal variability, reasonable agreement was found with observations from a ScanFish section, but only for the mesoscale wavenumber band; the observed sub-mesoscale variability was not reproduced by ROMS.

## 1 Introduction

In ocean acoustics research, the diagnostics and prediction of selected mixed-layer properties – such as the mixed-layer depth and the mixed-layer temperature – is of primary interest because they have a profound impact on the propagation of sound in the ocean. In this article, a high-resolution ocean circulation numerical model is presented which provides nowcasts and forecasts of these properties. The objectives are (i) to evaluate the sensitivity of the properties to different setups of initial conditions, lateral boundary conditions, atmospheric forcing patterns, the vertical grid, and vertical mixing parameterisations, and (ii) to find a setup which reproduces and predicts best the depth and the temperature of the mixed-layer and their associated spatio-temporal variabilities obtained from observations.

By definition, temperature and salinity are constant in the mixed-layer and the sound speed increases slightly with depth due to the pressure effect (Dietrich et al., 1975). Therefore, sound rays in the mixed-layer are refracted upwards and reflected at the sea surface. Hence, the mixed-layer acts as a *surface duct* (Katsnelson et al., 2012). On the other hand, at depth greater than the mixed-layer depth and because of the decreasing temperature, the rays are refracted in the other direction, i.e. towards greater depths. Consequently in terms of passive acoustic monitoring, if a sound source is within the mixed-layer, the sound cannot be "heard" at depths greater than the mixed-layer depth, and if the sound source is located below the mixed-layer, it cannot be heard in the mixed-layer. The equivalent is true for the location of objects by active sonar – if the sonar is within the mixed-layer, the acoustic signal can hardly reach an object at greater depth, and vice versa. This is, of course, an idealised model based on ray theory which does not take account of non-linear and frequency dependent effects, but it clearly emphasises that knowledge about the depth of the mixed-layer is mandatory for the planning and conduction of acoustic experiments.

The sound speed $c$ in seawater is a function of temperature $T$, salinity $S$, and the pressure $p$. Hence, small changes $dc$ of the sound speed can be described by the total differential

$$dc = \left.\frac{\partial c}{\partial T}\right|_{S_0,p_0} dT + \left.\frac{\partial c}{\partial S}\right|_{T_0,p_0} dS + \left.\frac{\partial c}{\partial p}\right|_{T_0,S_0} dp \tag{1}$$

where the subscripts $T_0$, $S_0$, $p_0$ indicate that $T$, $S$, and $p$, respectively, are held constant during the execution of the partial differential. For mid-latitudes and close to the sea surface ($T_0 = 15°C$, $S_0 = 35$, $p_0 = 0$ dbar), the partial differentials in Eq. (1) yield $\partial c/\partial T \approx 3.2$ m s$^{-1}$°C$^{-1}$, $\partial c/\partial S \approx 1.2$ m s$^{-1}$ which means that the fractional change of the sound speed with temperature is about three times larger than the change with salinity (Chen and Millero, 1977). Moreover, as typical spatio-temporal variations of temperature are $\mathcal{O}(10°C)$ but those of salinity only $\mathcal{O}(1)$ at best, the first two terms in Eq. (1) yield as 31.2 m s$^{-1}$ and 1.2 m s$^{-1}$. Hence, changes of the sound speed are largely controlled by changes of the temperature, and the impact of salinity variations in the mixed-layer can confidently be ignored for the calculation of the sound speed. However, one may note that this is only true for the open ocean. In coastal areas, estuaries, and in polar regions, the salinity variations are frequently larger and the concurrent variations of temperature smaller.

Besides the temperature and the depth of the mixed-layer, the temporal and horizontal variability of these two quantities require special attention (Pace and Jensen, 2002). The temporal variability at a fixed location is affected by temporal changes of

- Air-sea fluxes of momentum, heat, and fresh water,
- Activities of surface waves and internal waves,
- Horizontal advection,
- Vertical motion, and
- Optical properties of the seawater.

The horizontal variability is due to spatial differences of the same quantities and in addition to the presence of mesoscale and sub-mesoscale features like fronts, meanders, eddies, and filaments (e.g. Medwin and Clay (1998)). Both the temporal and the horizontal variability are impacting the sound speed and the underwater sound propagation.

The objective of this article is to find a model setup which predicts in the best possible way the mixed-layer properties and their spatio-temporal variabilities. While for the temporal variabilities the main focus of attention is directed at time scales between $\mathcal{O}(1\ \text{hour})$ and $\mathcal{O}(10\ \text{days})$, it is the intention to resolve the horizontal variabilities on scales on the order of 10 km and below. This requires a circulation model with a built-in vertical mixing scheme that accurately reproduces the diurnal cycle.

A state-of-the-art scheme has recently been published by Ling et al. (2015). It is an enhancement of the turbulence closure model of Noh et al. (2011) which is similar to Mellor and Yamada (1982) but in addition taking into account the effects of wave breaking and Langmuir circulation. Ling et al. (2015) developed new numerical techniques and improved schemes for the physics in Noh's model which amongst others intensified the diurnal amplitude of the simulated sea surface temperature. Noh et al. (2016) incorporated Noh's model in a global ocean general circulation model, and they could show that the new

mixing scheme helped to correct too high mixed-layer temperatures and too shallow mixed-layer depths in the high-latitude ocean. A similar approach was persued by a series of papers of Bernie et al. (2005, 2007, 2008): in the first publication, a one-dimensional mixed-layer model was developed based on the K-profile parameterisation of Large et al. (1994). The model was forced with observed fluxes from a mooring in the tropical Pacific Ocean, and it reproduced qualitatively the observed diurnal variability of the sea surface temperature over a period of about four months. However, for most of the time, the modeled

temperature was higher than the observed one, partly up to 1°C. Bernie et al. (2007) implemented the turbulent kinetic energy scheme of Blanke and Delecluse (1993) in an ocean circulation model, and finally, this circulation model was coupled with an atmospheric circulation model (Bernie et al., 2008). The major outcome of the latter publication was that the inclusion of the diurnal cycle leads to a tropic wide increase of the mean sea surface temperature, and in addition, the authors could demonstrate that the modeled diurnal cycle was modulated by intraseasonal variations. The vertical mixing in all papers mentioned above

was accomplished by turbulence closure models. By contrast, Gentemann et al. (2009) improved the parameterisation of the absorption of solar radiation in the diurnal heating bulk model of Fairall et al. (1996). This change, combined with a reduction of accumulated heat and momentum, increased the model's responsiveness to changes in surface heat flux and surface stress. Amongst others, the improved model predicted the vertical temperature profile within the diurnal thermocline, and increased warming at low wind speeds and decreased warming at high wind speeds.

The experimental area addressed in this study is situated in the Mediterranean Sea to the west of Sardinia (see Figs. 3, 4, 5, 7). Here, an oceanographic survey took place in June 2014 (Onken et al., 2014, 2016) the observational data of which are used to drive the aforementioned ocean circulation model and to validate the model results.

The reader may note that within this article all dates refer to the year 2014 and all times are in UTC (*Universal Time Coordinated*) except where otherwise stated.

## 2 Observational data

The observational data originate from the REP14-MED experiment which took place in the eastern Sardo-Balearic Sea in the period 6–25 June 2014. The collection of in situ data was accomplished by the NATO Research Vessel *Alliance*, the Research Vessel *Planet* of the German Ministry of Defence, a fleet of underwater gliders, surface drifters, one sub-surface float, and six

oceanographic moorings. Throughout this article however, the author will refer only to the data of one mooring denoted as *M1*, and to the CTD (*Conductivity-Temperature-Depth*) data collected by the survey vessels and the gliders. For more details of the observations, see Onken et al. (2016).

## 2.1 Mooring M1

M1 (Fig. 1) was launched on 8 June at 7:14 at 8°12.98' E, 39°30.80' N (Fig. 3) and recovered on 20 June at 13:55. The water depth at the launch position was ≈150 m. M1 consisted of the central mooring $M1_{CTR}$ and a sideways extending appendix $M1_{APP}$ floating at the sea surface. $M1_{CTR}$ was equipped with an upward-looking ADCP (*Acoustic Doppler Current Profiler*) mounted at a nominal depth of 100 m below the sea surface, a CTD probe at 1 m depth, and with a meteorological buoy at the surface. The appendix was connected by a 50 m long rope to $M1_{CTR}$ and extended to about 40 m in the vertical direction. 40 Starmon thermistors were mounted along the vertical cable to record the temperature with high vertical resolution. In addition, four RBR data loggers determined the actual depth of the Starmons. The nominal and actual vertical positions and the recorded parameters of the sensors are summarised in Table 1.

The Starmons recorded time $t$ and temperature $T$, and the RBRs in addition pressure $p$, both in intervals of about 10 s. The depth $z$ was calculated internally by the RBR software. As the Starmons did not have a pressure gauge, their actual vertical position was evaluated thereafter from the depth records of the RBRs because the positions of the Starmons relative to the RBRs was known. This was accomplished in the following way:

- The Starmon records at 3.0 m and 7.0 m depth were rejected because the recorders failed (cf. sensor positions 7 and 15 in Table 1).
- All records before 8 June 07:18 and after 20 June 13:30 were clipped, and then sub-sampled in 5-minutes intervals.
- At each instant, a second order polynomial fit was applied to the actual depth of the RBRs versus their nominal depth.
- The actual depths of the Starmon recorders were calculated with the polynomial using the previously calculated coefficients.

This procedure was advisable in order to correct for potential depth changes of the Starmons due to the actions of waves, internal waves, horizontal advection, and vertical shear. However, it turned out that these corrections were rather small: right at the sea surface at sensor position 1, the actual depth of the Starmon (cf. Table 1) varied between 0 m and 1.85 m, and the depth of the sensors at position 41 varied between 39.89 m and 41.81 m. Hence, the applied corrections were around ±1 m.

The time series of the recorded temperature and the vertical temperature gradient are shown in Fig. 2, several properties of which may be challenging to be reproduced by the circulation model:

- The near-surface temperature varied between about 22°C and more than 24°C. At the beginning of the recording period, it was around 23°C, then it rose slowly and reached the maximum value on 12 and 14 June, and afterwards it decreased. The minimum around 22°C was reached on 17 June, and during the final three days, the temperature rose again by about 1°C.
- The mixed-layer depth may be defined approximately by means of that depth where the vertical temperature gradient is maximal. Between about 15 and 20 June, there is clear evidence for such a signal – the mixed-layer depth varied between

about 4 m on 15 June and about 13 m on 18 and 19 June. However, between 8 and 14 June, the signal is rather indistinct: the night-time mixed-layer depth ranged from about 2 m on 9 June to about 6 m on 10 June, but during daylight hours the maximum gradient was sometimes found right at the surface; thus a mixed-layer in the "classical" sense did not exist.

- There are clear signals of the temporal variability, both for temperature and the depth of the mixed-layer. The variability occurred on all scales between about two weeks and the Nyquist period of 10 minutes (twice the sampling interval; see above).

## 2.2 Data collected by lowered CTD, gliders and towed measuring systems

On both survey vessels, casts with lowered CTD probes were conducted during the entire survey, but here only those casts taken during the 7–11 June period were used (for a more detailed description of the probes and their calibration, see Onken et al. (2016)). These casts belonged to the initialisation survey, the purpose of which was to provide realistic temperature and salinity data for the initialisation of numerical models. In total, 108 casts were taken on a regular horizontal grid with a mesh size of ≈10 km (Fig. 3), resolving the internal Rossby radius of deformation, the first mode of which is around 13 km (Grilli and Pinardi, 1998). Eleven gliders (for their payloads, see Onken et al. (2016)) were deployed on 8 and 9 June, respectively, at the positions marked "L" in Fig. 4, and afterwards directed to their nominal zonal tracks. The scheduled tracks were arranged parallel to the CTD sections (Fig. 3) but offset by 5 km in the meridional direction. For the validation of model forecasts, *Alliance* conducted a survey 21–24 June with a ScanFish. The tracks are shown in Fig. 5.

## 3 The circulation model

### 3.1 ROMS

The employed numerical ocean circulation model is ROMS, the *Regional Ocean Modeling System*. ROMS is a hydrostatic, free-surface, primitive equations ocean model, the algorithms of which are described in detail in Shchepetkin and McWilliams (2003, 2005). In the vertical, the primitive equations are discretised over variable topography using stretched terrain-following coordinates, so-called *s-coordinates* (Song and Haidvogel, 1994). In the version used for this article, spherical coordinates on a staggered Arakawa C-grid are applied in the horizontal, and the horizontal mixing of momentum and tracers is along isopycnic surfaces. The vertical mixing is parameterised by means of the GLS (*Generic Length Scale*) scheme developed by Umlauf and Burchard (2003). The air-sea interaction boundary layer in ROMS is based on the bulk parameterisation of Fairall et al. (1996b).

### 3.2 The model domain, discretisation

The model domain is situated to the west of Sardinia and it is identical to the area shown in (Fig. 3). The west and east boundaries are at 6°30.5' E and 8°35.5' E, while in the south and north the domain is limited by the 38°36.4' N and 40°59.6' N latitude circles, respectively. In east-west direction, the domain is separated in 120 grid cells, and in south-north direction in

178 cells, which yields an average grid spacing of $\Delta x \approx \Delta y \approx 1500$ m in the zonal and meridional direction, respectively. A comparison with Fig. 3 reveals that the domain boundaries are kept away from the observations by about 30 arc minutes; this was intended in order to mitigate a deterioration of the model solution at the observational sites due to false advection from the open boundaries.

Bathymetry data from the *General Bathymetric Chart of the Oceans* (GEBCO) with a spatial resolution of one arc minute were provided by the *British Oceanographic Data Centre* (BODC), and coastline data were obtained from NOAA (*National Oceanic and Atmospheric Administration*). In order to avoid crowding of the s-coordinates in shallow water regions, the bathymetry was clipped at 20 m which is the minimum allowed water depth, and for the smoothing of the bathymetry, a second-order Shapiro filter was applied. After smoothing, the so-called $rx0$ parameter resulted as 0.31 which is about 50%

higher than the maximum value of 0.2 recommended by Haidvogel et al. (2000), but $rx0$ is still less than 0.4 as suggested in the ROMS forum (`https://www.myroms.org/forum`).

    In the vertical direction, the model domain was separated in 70 s-layers, the position of which is controlled by three parameters $(\theta_s, \theta_b, h_c)$ and two functions, $V_{tr}, V_{str}$. Here, $V_{tr}$ is the *transformation equation*, $V_{str}$ the *vertical stretching function*, $\theta_s$ and $\theta_b$ are the *surface and bottom control parameters*, and $h_c$ is the *critical depth* controlling the stretching (for more details,

see `https://www.myroms.org/wiki/`). For all ROMS runs shown below, the functions and parameters were selected as $V_{tr} = 2$, $V_{str} = 1$, $\theta_s = 5$, $\theta_b = 0.4$, while $h_c$ was kept variable.

### 3.3   Initialisation

ROMS was initialised from nowcasts of the coarser *Mediterranean Forecasting System* (MFS, Dobrowsky et al. (2009); Tonani et al. (2014)) or the MERCATOR global ocean circulation model (Drévillon et al., 2008). In either case, the downscaling from the

20 *parent* to the *child* was accomplished first by linear horizontal interpolation of the prognostic fields on the ROMS grid. As the maximum horizontal resolution of MFS was close to 7 km (1/16°) and that of MERCATOR was 9.25 km (1/12°), the scale factors were around 4.7 and 6.2, respectively. Thereafter, all fields were interpolated vertically from the horizontal depth levels to the s-coordinates. A special issue was the alignment of the land masks of the parent and the child: if any wet grid cell of the child was covered by a dry grid cell of the parent, a smooth transition of all variables was created by taking the average of the

25 surrounding parent cells. However, as this may lead to a violation of continuity by non-zero horizontal velocities normal to the land mask, all horizontal velocities next to the ROMS land mask were set to zero.

### 3.4   Lateral boundary conditions and nesting

The ROMS code includes various methods for the treatment of open boundaries. After extensive sensitivity studies, it was found that the following algorithms served best for the posed problem: for the sea surface elevation, the *Chapman condition*

was selected (Chapman, 1985), and for all other quantities (i.e. barotropic and baroclinic momentum, turbulent kinetic energy, temperature, and salinity), the *mixed radiation-nudging conditions* after Marchesiello et al. (2001) were applied.

    The lateral time-dependent boundary conditions were provided by the parent by means of one-way nesting. However, the information from the parent was not instantaneously superimposed to the ROMS solution but an additional nudging was applied

to all prognostic variables (except for the sea surface elevation) which allowed these fields to adjust slowly to the parent values at the boundaries within an e-folding time scale of two days. In addition, a factor 5 was used for the nudging time scales which caused a stronger nudging on the inflow.

## 3.5 Surface boundary conditions

At the sea surface, boundary conditions for the air-sea exchange of fresh water, momentum, and heat were evaluated from the outputs of two numerical weather prediction models and from the measurements of the meteorological buoy on top of M1 (see Fig. 1) by means of the wind field at 10 m (2 m for M1) height, air temperature and relative humidity at 2 m, air pressure at sea level, cloudiness (not available from M1), short wave radiation, and precipitation. The output of the weather prediction models was made available by the Italian Weather Service CNMCA (*Centro Nazionale di Meteorologia e Climatologia Aeronautica*)

in two different setups, COSMO-ME and COSMO-IT. COSMO-ME covers the entire Mediterranean Sea with a horizontal resolution of 7 km and provided 72-hours forecasts, while COSMO-IT encompasses Italy and the adjacent waters at the very high resolution of 2.2 km but the forecast range was only 24 hours. The temporal resolution of both models was 1 hour. The time series of all available variables from COSMO-ME, COSMO-IT, and the meteorological buoy are shown in Fig. 6 at the M1 position.

## 3.6 Data assimilation

In most of the model runs presented below, temperature and salinity data from shipborne CTD probes and gliders were assimilated using *Objective Analysis* (OA, see Bretherton et al. (1976); Carter and Robinson (1987); Thomson and Emery (2014)). Namely, ROMS includes a module which enables data assimilation by the *4D-Var* method. However, as 4D-Var is based on variational methods it is rather expensive in terms of computer resources; according to parallel ROMS runs but using 4D-Var

(A. Funk, personal communication, 2016), the CPU time increases by about a factor of 10 compared to OA. During the integration of ROMS, the engine conducting the data assimilation was invoked every day at midnight, and it was controlled by six parameters:

- $W$: the width of the time window which determines what data are assimilated. In all ROMS runs below, $W = 48$ hours; this setting was found to provide the best forecast skill (Onken, 2017). Hence, all temperature and salinity data of the

previous and the following 24 hours were selected for assimilation.

- $C$: the isotropic correlation length scale. $C = 15$ km was used throughout which is approximately the internal Rossby radius of the Western Mediterranean in summer (Grilli and Pinardi, 1998). Isotropic correlation is a strong assumption especially close to the coast. However, according to the observations from ADCP measurements (I. Borrione, personal communication, 2016), predominantly meridional currents were prevailing only in a 10-km wide stripe along the Sar-

dinian coast while the rest of the 180-km wide model domain was characterised by an eddy field with alternating currents. Here, the usage of a non-isotropic correlation scale would deteriorate the results.

- $\delta T_{obs}$, $\delta S_{obs}$: the *observational errors* of temperature and salinity, respectively. $\delta T_{obs} = 0.5°C$ and $\delta S_{obs} = 0.16$ were used throughout. These values were obtained from the variance of all CTD casts in the upper thermocline.

- $\delta T_{clim}$, $\delta S_{clim}$: the *climatology errors*; $\delta T_{clim} = 5 \times \delta T_{obs} = 2.5°C$ and $\delta S_{clim} = 5 \times \delta S_{obs} = 0.8$ were applied.

## 3.7   Integration and output

All ROMS runs presented below were initialised on 1 June 00:00 and integrated forward for 24 days until 25 June 00:00. To satisfy the horizontal and the vertical CFL criterion, a baroclinic time step $\Delta t = 108$ s (800 steps per day) was chosen, and the number of barotropic time steps between each baroclinic time step was 40. Harmonic mixing along isopycnals with an eddy diffusivity coefficient of 5 m²s⁻¹ was used for the horizontal diffusion of the tracers $T$ and $S$, and a horizontal viscosity coefficient of 1 m²s⁻¹ was selected for the diffusion of momentum. Further on, a quadratic law using a coefficient of 0.003 was

applied for the bottom friction, and the pressure gradient term was computed using the standard density Jacobian algorithm of Shchepetkin and Williams (2001, unpublished; see

`http://www.atmos.ucla.edu/~alex/ROMS/pgf1A.ps`).

The three-dimensional volume of all prognostic fields was written to an output file at 6-hours intervals. For comparison of the ROMS results with the observed records at mooring M1, time series of vertical temperature profiles right at the position of

M1 where written to an extra file at the full temporal resolution.

## 4   Sensitivity of near-surface temperature and mixed-layer depth

The purpose of this section is to investigate the impacts of:

- initialising ROMS from different data sets,
- the setup of the vertical grid,

- different atmospheric forcing patterns,
- different vertical mixing schemes,
- the background eddy diffusivity,

on the temperature between the surface and about 42-m depth (which was the vertical range of the M1 observations) and the depth of the mixed-layer. This was achieved by five series of ROMS runs named A–E (see Table 2 for the parameter settings

and results of each model run), including 28 runs in total. The task of each series was to assess the sensitivity of the ROMS forecast skill to variations of the mechanisms mentioned in the bullets above. For each run, the ability of ROMS to predict the temperature was assessed by means of the root-mean-square (*rms*) difference

$$\Delta T = \left[ \frac{1}{N} \sum_{1}^{N} (T_{ROMS} - T_{obs})^2 \right]^{\frac{1}{2}} \tag{2}$$

between the observed temperature $T_{obs}$ and the predicted temperature $T_{ROMS}$ at each depth level of the observations. $\Delta T$ was

evaluated for the period 15 June 00:00–20 June 13:55 where $N$ observations were available in 5-minute intervals (cf. Section

2.1). This interval was selected because it enabled the comparison of all runs with those which were forced by data assimilation until 12 June 00:00. The three-days lag between the last assimilation on 12 June and the start of the evaluation period on 15 June was granted to ROMS in order to recover from "assimilation shocks" which frequently become noticeable in the form of strong inertial oscillations. The experience from precursor model runs has shown that such oscillations die off after about 3–4 inertial periods (18.7 hours at 40°N). In order to synchronise the modelled and the observed temperature, $T_{ROMS}$ was linearly interpolated in space and time on the observations. The equivalent method was also applied to the mixed-layer depth, $D$, which due to the lack of salinity observations at the M1 position was defined as that depth where the temperature was for the first time by 1°C colder than the temperature at the surface (cf. Lamb (1984); Wagner (1996)), hence

$$\Delta D = \left[ \frac{1}{N} \sum_{1}^{N} \left( D_{ROMS} - D_{obs} \right)^2 \right]^{\frac{1}{2}} \tag{3}$$

is the *rms* difference of the mixed-layer depths.

## 4.1 Series A: initialising ROMS from different data sets

In this series, $h_c = 10$ m was selected for the critical depth. In the first run, referred to as A1, ROMS was initialised from MFS while in A2, the initial conditions were provided by MERCATOR. A3 was initialised from MERCATOR as well but in addition, temperature and salinity data from CTD casts and 10 gliders were assimilated which were taken 7–12 June 00:00 (cf. Section 2.2 and Figs. 3, 4, 7). The surface boundary conditions of all runs in the A-series were provided by COSMO-ME.

In Fig. 8a are shown time series of the near surface temperatures at 0.81 m depth from runs A1–A3, in comparison with the corresponding observations of the uppermost Starmon sensor in M1 at the same depth level. In A1 and A2, the predicted temperatures agree reasonably well with the observations after 15 June but before, the temperature exceeds the observations by several degrees. Extreme differences are visible 12–14 June with differences of close to 3°C. Fig. 6 shows that during this period the predicted and observed wind speeds were close to 0 m s$^{-1}$ and the short-wave radiation flux reached maximum values of more than 800 W m$^{-2}$. Hence, as these quantities are the major drivers of the mixed-layer temperature, it is concluded that the parameterisation of the vertical mixing in ROMS is not adequate for such calm situations. By contrast, as soon as the wind became stronger after 14 June, the maximum difference between the predicted and measured temperature is less than 1°C. In A3 before 12 June, there is a better agreement between the modelled and the observed temperature. However, as can be seen from the sudden drop of the modelled temperature at midnight on 10–12 June, the data assimilation led to an underestimate of the surface temperature. The reasons for this are twofold: first, some of the assimilated profiles started at 2 m or even 3 m depth because the measurements close to the surface were not reliable. In such cases, the uppermost measurements were extended to the surface and led to an underestimate of the near-surface temperature, which sometimes was significant because of the extremely shallow or even non-existent mixed-layer. And second, the OA "advected" properties from neighbouring casts which were not representative for the M1 position. On 13 June, the modelled temperature exceeds again the observations by almost 2°C but the difference is less than in A1 and A2. And after 15 June, the A3 temperature is rather close to the temperatures

in A1 and A2. As a skill measure for the forecasted near-surface temperature, $\Delta T$ was evaluated for all runs and resulted as $\Delta T = 0.30°$C in A1, $\Delta T = 0.53°$C in A2, and $\Delta T = 0.51°$C in A3 (see also the legend box in Fig. 8 and Table 2).

The temporal evolution of the mixed-layer depth is displayed in Fig. 8b. As revealed by the decreasing *rms* differences $\Delta D$ between the modelled and observed mixed-layer depths, the forecast skill increases from A1 to A2 and from A2 to A3.

Noteworthy is the close agreement between the observed and modelled mixed-layer depth in A3 before 12 June which was forced by the assimilation. Remarkable is again the mismatch between the model and the observations 12–15 June which is another indication for an inadequate parameterisation of the mixed-layer dynamics at low wind speeds.

The vertical distribution of the *rms* temperature differences $\Delta T$ of all runs of the A-series is shown in Fig. 9. It is demonstrated that at most depth levels, $\Delta T$ is lower or equal in A2 compared to A1. The assimilation in A3 led to a further significant

decrease between about 4 m and 35 m depth; only above 4 m and below 35 m depth, $\Delta T$ is higher in A3. The general better forecast skill of A3 is also supported by $\overline{\Delta T}$, the vertical mean of $\Delta T$, which is greater than 1°C in A1 and A2 but only 0.90°C in A3 (see also Table 2). In summary, nesting ROMS in MERCATOR and assimilating CTD profiles provided the best forecasts for the temperature and the depth of the mixed-layer and the thermocline temperature below about 4 m depth. Therefore, all runs in the B-series will be based on A3.

The temporal evolution of the modelled temperature in A3 at the position of mooring M1 is shown in Fig. 10b. In comparison with Fig. 10a, the modelled temperature close to the sea surface is too high on 13 and 14 June, while at depths greater than about 3 m–10 m, $T_{A3}$ appears too low. This is confirmed by Fig. 10c which exhibits the temperature difference $T_{A3} - T_{M1}$: in approximately the upper 2-m depth range, $T_{A3}$ partly exceeds $T_{M1}$ by about 2°C during these days, and just below, $T_{A3}$ is up to more than 3°C lower than the observed temperature. This aberrant cold layer can be identified during the whole

model run. Apparently, the modelled mixed-layer depth is shallower than the observed one. This is illustrated by the vertical temperature gradient in Fig. 10e. Namely, a comparison with Fig. 10d reveals that the general descending trend of the maximum gradient is similar, but the depth of the modelled maximum is always less than the observed one. Moreover, the observed variability is significantly higher than the modelled one. While for the entire period, there is clear evidence for a strong diurnal variability in the observations (e.g. the deep mixed-layer in the early morning and the shallow mixed-layer in the afternoon),

the modelled variability is much less pronounced. Another feature worth mentioning is that the thermocline is too warm during the assimilation phase before 12 June (Fig. 10c). It has been verified that this was caused by the assimilation of glider data because this feature is not present in a run where only casts from lowered CTD were assimilated (not shown). As can be seen from Figs. 3 and 7, two CTD casts were taken exactly at the M1 position while numerous glider casts are close to M1 (note that the meridional offset of the glider tracks with respect to the CTD meridional sections was 5 km). Thus, as the correlation

scale of the OA was 15 km, the modelled temperature at M1 was primarily determined by the glider measurements because the large amount of glider profiles reduced the statistical weight of the two CTD casts.

## 4.2 Series B: sensitivity to the setup of the vertical grid

If the transformation equation, the vertical stretching function, and the total number of layers are held constant, the layer thicknesses of the ROMS vertical grid is controlled by the surface and bottom control parameters, $\theta_s$ and $\theta_B$ and the critical depth,

$h_c$. For mixed-layer modelling in shelf areas, it would be desirable to have high vertical resolution close to the surface which can be achieved by either increasing $\theta_s$ or decreasing $h_c$. However, as increasing $\theta_s$ would make the vertical transformation more non-linear, it was decided to keep $\theta_s = 5$ constant and vary only $h_c$. In this series, the sensitivity of the ROMS results to five different settings of the critical depth is investigated, using $h_c \in \{10, 20, 50, 100, 200\}$ m. For each of these choices, the

impact on the layer thicknesses at the position of mooring M1 is illustrated in Fig. 11. A minimum layer thickness of 0.27 m right at the sea surface is achieved by $h_c = 10$ m in Run B1, while the thickness of that layer gradually increases in B2–B5. In the latter ($h_c = 200$ m), the thickness is close to 1.3 m. B1 being identical to A3 is the control run.

For all runs of this series, the temporal evolution of the mixed-layer properties is displayed in Fig. 12. Although still too high around 14 June, the near-surface temperatures in all runs of this series are most resembling the observations during the entire

integration period which is also expressed by the corresponding low values for $\Delta T$; the minimum $\Delta T = 0.44°$C is obtained from B5, while the highest is found in B1 ($\Delta T = 0.51°$C). For the mixed-layer depth, there is no clear evidence which run might do best. $\Delta D$ varies only in a rather narrow range between 2.62 m in B1 and 2.75 m in B5. The vertical distributions of $\Delta T$ (Fig. 13) and as well the vertical averages $\overline{\Delta T}$ are almost identical for all runs but right at the surface, $\Delta T$ is minimal in B5 as shown already in Fig. 12a. As the above results did not reveal a clear tendency which choice of $h_c$ yielded the best results, it

was decided to continue with B1 ($h_c = 10$ m) as control run in Series C below. This decision was guided by Bernie et al. (2008) who asserted that a minimum vertical resolution of 1 m is mandatory to resolve the diurnal cycle of the sea surface temperature. Another criterion for this decision was the $rx1$ grid parameter being minimum in B1 (see Table 2). Namely, according to the ROMS discussion forum (https://www.myroms.org/forum), $rx1 > 14$ is considered as "insane" because the Haney (1991) condition is violated. However, there are various contributions in the forum, reporting that even with $rx1 >> 14$ there

did not arise any problems with the corresponding ROMS runs.

## 4.3   Series C: sensitivity to atmospheric forcing

Series C consists of three model runs, C1, C2, and C3. C1 is identical to B2; in C2, the surface boundary conditions were provided by COSMO-IT instead of COSMO-ME. And in C3, the atmospheric forcing was defined by means of the observations of the meteorological buoy on top of mooring M1. Here, the observations were spread uniformly across the entire model domain

whenever available. If no observations were available, i.e. before 8 June and after 20 June, the atmospheric fields of COSMO-ME were used. As observations of cloudiness were not available from M1, the corresponding fields from COSMO-ME were used throughout.

According to Fig. 14a, the predicted near-surface temperature from C2 resembles closely that of C1, except for 14–17 June where the temperatures in C2 are about 1°C higher. Apparently, this was driven by different wind forecasts of the weather

prediction models (cf. Fig. 6). Before 14 June, the wind forecasts of both models were almost identical, but for the following two days during a period of stronger winds, the forecasts differ from each other. Overall, the near-surface temperature does not appear to be very sensitive to the choice of the weather forecast models. This is also expressed by $\Delta T$ which attains similar values of 0.51°C and 0.42°C, respectively. The signature of the temperature changes considerably when ROMS was driven by the weather observed at M1: this is already evident 8–10 June where the modelled temperature in C3 is different from C1 and

C2. And after 15 June, it is mostly higher than both the observations and the predictions of C1 and C2, which correspondingly leads to a higher $\Delta T$ of 0.80°C. With respect to the modelled mixed-layer depth (Fig. 14b) and based on the $\Delta D$ criterion, C1 is superior to C2 and C3, but the large discrepancies 12–15 June between predictions and observation are still present in all three runs. This corroborates the above hypothesis that the mismatch is not caused by the atmospheric forcing because the

most appropriate forcing was applied in C3.

     A surprising result was obtained from the vertical structure of the *rms* temperature difference (Fig. 15). Below about 3 m depth, $\Delta T_{C1}$ is about 0.1°C less than $\Delta T_{C2}$ but a considerable improvement of the predicted stratification is provided by C3. In the entire vertical range below about 5 m, $\Delta T_{C3}$ is up to 0.4°C less than $\Delta T_{C1}$. Only right at the surface, $\Delta T_{C3}$ is approximately 0.3°C higher than the corresponding values from C1 and C2 which is obviously due to the above mentioned

mismatch after 15 June. C3 provides the best results for the temperature stratification in the thermocline. As the temperature in this depth range was definitely not affected by the heat exchange at the sea surface ($\approx$90% of the short-wave radiation is absorbed in the uppermost 1 m depth range), its improvement could only be achieved by lateral advection which is controlled by the wind; apparently, the wind is better represented in the observations than in the weather forecasts. Summarised, the objective skill measure $\Delta T$ for the near-surface temperature and $\Delta D$ for the mixed-layer depths indicate that C1 provides the

best forecast, while $\Delta T_{C3}$ is clearly superior to $\Delta T_{C1}$ and $\Delta T_{C2}$ in the thermocline. The latter consolidates the decision to use C3 as control run in Series D because advective processes are obviously reproduced best.

     The decision for C3 is supported by Fig. 16: by visual inspection, the evolution of the predicted temperature pattern in C3 (Fig. 16c) resembles the observations (Fig. 16a) more than in C1 (Fig. 16b). Namely, the near surface temperature is too high but the thickness of the warm layer 16–20 June is roughly the same as in the observations, close to 10 m. Moreover, the depth

and the variability of the maximum vertical temperature gradient in C3 resembles to a larger degree the observed pattern (cf. Fig. 16d, e, f) although the vertical temperature gradient is still too weak.

### 4.4   Series D: sensitivity to the vertical mixing parameterisation

The GLS scheme (Umlauf and Burchard, 2003) provides a generalisation of a class of differential length-scale equations used in turbulence models for oceanic flows. Commonly used models, like the k-kl model of Mellor and Yamada (1982), the k-$\epsilon$

model (Rodi, 1987), and the k-$\omega$ model (Wilcox, 1988) are recovered as special cases of the generic scheme. Here, k is the turbulent kinetic energy, l the length scale of the turbulence, $\epsilon$ the dissipation rate, and $\omega$ the specific dissipation rate. In Series A–C, the GLS vertical mixing scheme was applied using its generic parameters as formulated by Umlauf and Burchard (2003). In the following, D1 is identical to C3, serving as control run, in D2 is applied the GLS scheme with the k-kl parameterisation, in D3 the k-$\epsilon$ parameters, and finally in D4 the k-$\omega$ parameterisation which was adjusted to oceanic conditions by Umlauf et al.

(2003).

     After 12 June, the near-surface temperature of all runs is correlated with the observations (Fig. 17a), but mostly it is still too high. Moreover, the graphs indicate that the temperatures from D2, D3, D4 are closer to the observed ones which is also expressed by $\Delta T_{D2} = 0.50$°C, $\Delta T_{D3} = 0.51$°C, $\Delta T_{D4} = 0.41$°C while $\Delta T_{D1} = 0.80$°C. For the mixed-layer depth (Fig. 17b), the best agreement with the observations was obtained from D4 with $\Delta D_{D4} = 2.71$ m. However, the mixed-layer was

mostly too shallow in all runs of this series. Hence, based on the $\Delta T$ and $\Delta D$ criteria, the k-$\omega$ mixing scheme in D4 definitely performs best. This is also supported by the vertical structure of $\Delta T$ displayed in Fig. 18. There is clear evidence that the k-kl scheme (D2), the k-$\epsilon$ scheme (D3) and the k-$\omega$ scheme (D4) do better than the generic GLS (D1), but between the surface and about 5 m depth, the best result was obtained from D4. Therefore, D4 will serve as control run in the following E-Series.

5 An indicator why the k-$\omega$ parameterisation performed better than the other closure schemes is possibly found in the publication of Reffray et al. (2015). Here, a one-dimensional model implemented in a three-dimensional circulation model was used to investigate physical and numerical turbulent-mixing behaviour. Amongst others, the k-kl, the k-$\epsilon$, and the k-$\omega$ scheme were compared to each other. It turned out that the k-$\omega$ scheme was most sensitive to the vertical resolution: in a coarse (about 10 m) resolution model, k-kl and k-$\epsilon$ did clearly better than k-$\omega$ while at high (about 1 m) resolution, all three schemes yielded suitable results. In the D-Series, the vertical resolution close to the sea surface is 0.27 m (see Fig. 11 and Section 4.2 above). Hence, one may speculate that the k-$\omega$ formulation becomes superior to the other schemes when increasing the vertical resolution.

## 4.5 Series E: sensitivity to the background vertical eddy diffusivity

The shortcoming of all model runs conducted so far was that the mixed-layer was too warm and too shallow, and the thermocline was too cold with respect to the observational data. This is also in agreement with the findings of Reffray et al. (2015). Hence,
15 it was conjectured that the parameterisation of the vertical transport of heat and/or momentum was not adequate. Several attempts were undertaken to fine-tune the D4 results by varying the vertical eddy viscosity coefficient and the turbulent closure parameters, but the outcomes were sobering – a significant improvement of the forecast skills for the mixed-layer properties was not achieved. Hence, in this series, the background vertical eddy diffusivity $A_{VT}$ was increased gradually from $1 \times 10^{-6}$ m$^2$ s$^{-1}$ in E1 (which is the control run identical to D4) to $2 \times 10^{-4}$ m$^2$ s$^{-1}$ in E13. The forecast skill of each run was again
20 assessed by means of $\Delta T$ at 0.81 m depth, and by $\Delta D$. The dependency of these parameters on $A_{VT}$ are shown in Fig. 19. $\Delta T$ exhibits minimum values of $0.31°$C ($\approx 0.1°$C lower than in D4) for $A_{VT} \leq 2 \times 10^{-5} \leq 3 \times 10^{-5}$ m$^2$ s$^{-1}$ in E4 and E5 which is somewhat higher than $(1.7 \pm 0.2) \times 10^{-5}$ m$^2$ s$^{-1}$ obtained from tracer measurements in the thermocline during the North Atlantic Tracer Release Experiment (Ledwell et al., 1998; Thorpe, 2007). By contrast, the minimum of $\Delta D = 2.05$ m is found in E9 for $A_{VT} = 7 \times 10^{-5}$ m$^2$ s$^{-1}$.

25 In Fig. 20 are shown the near-surface temperature in E4 and the mixed-layer depth in E9, together with the corresponding quantities of the control run E1 and the observations. After 15 June, the increase of $A_{VT}$ from $1 \times 10^{-6}$ m$^2$ s$^{-1}$ to $2 \times 10^{-5}$ m$^2$ s$^{-1}$ shifted the near-surface temperature by about $0.1°$C closer to the observations. For most of the time, the modelled signal is correlated with the observations, although the modelled maximum and minimum temperatures are frequently lagged a few hours behind the observed extreme values. Similar features were also described by Gentemann et al. (2009) when comparing
30 time series of observed sea surface temperatures with those generated by the model of Fairall et al. (1996). In their improved model (see Introduction), they demonstrated that the peak warming in the afternoon was shifted earlier. For the mixed-layer depth, the increase of the eddy diffusivity to $7 \times 10^{-4}$ m$^2$ s$^{-1}$ caused a significant reduction of $\Delta D$ from 2.71 m in E1 to 2.05 m in E9. While in the precursor series the mixed-layer was always too shallow, it agrees now remarkably well with the observations, except for large discrepancies on 19 and 20 June where the predicted mixed-layer is up to 4 m shallower than the

observed one. As the M1 wind speed was very low during these days (Fig. 6), other processes leading to a deepening of the mixed-layer were probably inadequately parameterised, such as Langmuir circulation and wave breaking (cf. Noh et al. (2011, 2016)).

## 5 Temporal variability

In order to assess the modelled temporal variability of the temperature and the depth of the mixed-layer, the normalized spectra of the near-surface temperature amplitude $\hat{T}$ at 0.81 m depth and of the mixed-layer depth amplitude $\hat{D}$ were computed by Fourier transform both from the observations and the ROMS outputs of runs E4 and E9, respectively. To enable a sufficient spectral resolution for the cycle periods around 1 day, the entire time series between 8 and 20 June was used as input for the Fourier transform. At first glance, the modelled spectrum of the near-surface temperature (Fig. 21a1) resembles the observations

in the cycle period range between about 0.1 and 1 days but significant differences are evident in the bands between about 0.1 and 0.4 days where the modelled amplitude is partly up to one order of magnitude different from the observed one. This mismatch is not surprising, because here the temporal variability is controlled mainly by internal waves which are not or only partially reproduced by the model. By contrast, the 0.4–0.8 days band (10–19 hours) is the range dominated by tides and inertial motions. Theoretically at 40°N, the inertial peak is at 18.7 hours (0.78 days), but a corresponding small peak is only

visible in the modelled spectrum; no such peak is noticeable in the observations. Probably, the modelled peak is a leftover of the assimilation shock on 12 June. Additional peaks are found both in the modelled and the observed spectrum at about 0.4, 0.5, and 0.6–0.7 days ($\approx$10 hours, $\approx$12 hours, $\approx$14–17 hours). While the sources of the first and the latter are unknown, the 12-hours peak might be related to a semi-diurnal tidal component. However, as there was no tidal forcing in the ROMS version utilised in this study, and the MERCATOR forcing at the lateral boundaries was defined by means of daily averages,

the semi-diurnal variability could only be caused by tides of the atmosphere. Both the modelled and the observed spectrum are dominated by the diurnal variability, represented by the peak at 1.0 days. In the red part of the spectrum between 1 and about 10 days, the modelled and observed amplitudes exhibit some weak correlation and they are of about the same order of magnitude, but the author refrains from discussing this matter which is potentially impacted by long-period fluctuations of the forcing at the surface and at the lateral boundaries. More detailed information of the correlation $r_{\hat{T}_{ROMS}, \hat{T}_{obs}}$ between the modelled

and observed temperature amplitudes is gained from Fig. 21a2: the correlation coefficient $r = 0.74$ together with the p-value $p = 3.05 \times 10^{-22}$ proves a high significant correlation, and the regression coefficients $a_0 = 0, a_1 = 1.74$ indicate that in general the modelled amplitudes are overestimated. By contrast, there is less but still significant correlation between the modelled and the observed mixed-layer amplitudes $\hat{D}_{ROMS}$ and $\hat{D}_{obs}$, respectively (Fig. 21b2), which is indicated by $r_{\hat{D}_{ROMS}, \hat{D}_{obs}} = 0.50$ and $p = 4.52 \times 10^{-9}$. This finding is also supported by the spectrum (Fig. 21b1) where a slight correlation of the amplitudes is

only found for the diurnal and semi-diurnal cycles.

## 6 Horizontal variability

In order to assess the capability of ROMS to reproduce and predict the horizontal variability of mixed-layer properties, the results of Run E9 were analysed along the ScanFish tracks A03, A05, A07, A09, and A10 (see Fig. 5), and compared with the data collected by the towed device. E9, using $A_{VT} = 7 \times 10^{-5}$ m$^2$ s$^{-1}$, was selected for this comparison because both $\Delta T$ and $\Delta D$ were acceptable. Details of the ScanFish tracks are summarised in Table 3. As ROMS output was only available in 6-hours intervals starting at midnight, in each case that output cycle was used which fell within the time window when the tracks were conducted. This assumed synopticity of the ScanFish tracks which is justified by the fact that the maximum duration of the tracks was 5 hours, 28 minutes for A03.

To make the ScanFish observations and the ROMS products comparable, the ScanFish temperature was interpolated vertically on 1-dbar standard levels and the ROMS temperature was mapped on the same levels. As the upper inflection point of the ScanFish varied between about 5 and 10 dbar, there was frequently no information on the near-surface temperature available. In such cases, the temperature at the inflection level was extended to the surface. The same method was applied to the ROMS temperature which was not defined right at the surface but in the centre of the first s-layer below the surface. In deep-water regions, this was located at about 3-m depth.

In Fig. 22a is shown a temperature section from the ScanFish measurements along the central track A05, and the corresponding section from ROMS is displayed in Fig. 22b. The gross features of both sections resemble each other, but the small-scale horizontal variability of the Scanfish temperature was not reproduced by ROMS. This is probably due to the smoothing effect of the OA or due to the combined action of the horizontal eddy diffusivity and numerical diffusion. However, as the last assimilation cycle was conducted on 12 June ten days prior to the ScanFish observations, one may exclude that the OA removed the small scale features. Moreover, the vertical temperature gradient is much weaker in ROMS which was already noticed above. Hence, this is apparently not caused by the increased vertical diffusivity but by the vertical resolution of ROMS. The sea surface temperatures and the mixed-layer depths from the ScanFish and ROMS are displayed in Figs. 22c and 22d. For the surface temperature, the observed large-scale west-east trend is reproduced by ROMS but there are differences of up to about 0.5°C in the central portion of the section. Maximum differences between the modelled and the observed mixed-layer depth in the 0–20 km range are close to 5 m at 13 km distance, while in the eastern half of the section, the modelled and the observed mixed-layer depths are approaching each other. However, the smaller-scale $\mathcal{O}(1$ km$)$ observed variability was not reproduced by ROMS both for the sea surface temperature and the mixed-layer depth.

To investigate why the small-scale variability was not predicted correctly, run E9 was repeated but now using a smaller horizontal eddy diffusivity coefficient of 1 m$^2$ s$^{-1}$ instead of 5 m$^2$ s$^{-1}$ which was used for all model runs so far. However, no significant changes were noticeable. Thus, one has to settle for the fact that the present setup of ROMS is only able to reproduce the horizontal variability of mixed-layer properties on those scales which are comparable to the Rossby radius.

# 7 Conclusions

ROMS has been utilised to diagnose and predict properties of the ocean mixed-layer, and the sensitivity of the model results to the choice of the initial and boundary conditions, the setup of the vertical grid, and vertical mixing schemes was investigated. The initial and lateral boundary conditions for ROMS were taken from two different parent models by one-way nesting. At the surface, ROMS was forced by two different weather forecasts, or by observations. All ROMS nowcasts and forecasts were validated against observations which were taken in June 2014 to the west of Sardinia in the Mediterranean Sea.

To explore the sensitivity of the near-surface temperature and the mixed-layer depth to the choice of the initial conditions, ROMS was alternatively initialised by the *Mediterranean Forecasting System*, MFS, and the global MERCATOR model, and in addition, observed temperature and salinity data were assimilated. For validation, time series of temperature were compared with observations from a mooring. Initialising ROMS from MERCATOR instead of MFS provided a better agreement between the model and the observations, but a significant improvement was obtained from a ROMS run initialised from MERCATOR and updated with assimilated data from CTD casts and gliders. This applied both to the near-surface temperature, the mixed-layer depth, and as well to the temperature distribution in the upper thermocline.

To investigate the impact of the surface boundary conditions, atmospheric forcing fields were taken from the weather prediction models COSMO-ME and COSMO-IT, and from observations of a meteorological buoy acting as a point source. With respect to the mixed-layer depth, the best agreement with the observations was obtained from a model run forced with COSMO-ME, while the near-surface temperature exhibited the best match when ROMS was forced by COSMO-IT. However, the stratification in the upper thermocline was best represented when the point source was applied. The obvious reason for this surprising result is that the momentum forcing was overestimated by both COSMO-ME and COSMO-IT.

For the vertical mixing, four different configurations of the GLS scheme of Umlauf and Burchard (2003) were applied, representing the generic version, the k-kl model of Mellor and Yamada (1982), the $k$-$\epsilon$ model (Rodi, 1987), and the k-$\omega$ model (Wilcox, 1988). The best performance was obtained from the k-$\omega$ model.

Regardless of which initial conditions or surface boundary conditions were applied – the modelled mixed-layer was always too shallow and too warm. Therefore, the background vertical eddy diffusivity coefficient, $A_{VT}$, was varied over more than one order of magnitude. The best agreement of the mixed-layer temperature was obtained for $A_{VT} \approx 2 \times 10^{-5}$ m$^2$ s$^{-1}$ while $A_{VT} = 7 \times 10^{-5}$ m$^2$ s$^{-1}$ provided the best match of the mixed-layer depth with the observations.

A positive and significant correlation was found between the modelled and the observed temporal variability of the mixed-layer temperature. The modelled variability resembled the observed variability predominantly for cycle periods in the spectral ranges between about 0.5 and 1 days. By contrast, less correlation was found between the modelled and the observed variability of the mixed-layer depth. A slight agreement was only found for the diurnal period.

The horizontal variability was validated against measurements from a high-resolution zonal ScanFish section. Both the modelled mixed-layer temperature and the mixed-layer depth closely resembled the observations, but only on the larger scales of $\mathcal{O}(10$ km$)$. Hence, the mesoscale variability was rather well reproduced but the sub-mesoscale variability was not.

### Code availability

All work related to this article was done on a Linux workstation under Kubuntu 16.04. ROMS/TOMS version 3.6 was used for the model runs, the pre- and postprocessing was done with MATLAB R2016b, and the article was written in LaTeX. The model code and all scripts are available from the author on request.

### Data availability

All data of the REP14-MED experiment are available on the CMRE data server at `http://geos3.cmre.nato.int`. The data are NATO UNCLASSIFIED and available only for the partners of the experiment. However, interested institutions can sign up for partnership at any time. Requests may be directed to the author or to <geos-webmaster@nurc.nato.int>.

### Competing interests

The author declares that he has no conflict of interest.

*Acknowledgements.* The author would like to thank the masters and crews of NRV *Alliance* and RV *Planet* for their professionalism during the conduction of the experiments at sea. The data from COSMO-ME and COSMO-IT were provided by the Italian weather service *Centro Nazionale di Meteorologia e Climatologia Aeronautica*. REP14-MED was sponsored by *HQ Supreme Allied Command Transformation* (Norfolk, VA, USA).

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

**Table 1.** Nominal and actual depths of the Starmon and RBR sensors mounted on the appendix of mooring M1. For the meaning of the recorded variables see text.

| sensor position | sensor type | recorded variables | nominal depth [m] | mean actual depth [m] | remarks |
|---|---|---|---|---|---|
| 1 | Starmon | $t, T$ | 0.0 | 0.81 | |
| 2 | Starmon | $t, T$ | 0.5 | 1.30 | |
| 3 | Starmon | $t, T$ | 1.0 | 1.79 | |
| 4 | Starmon | $t, T$ | 1.5 | 2.28 | |
| 5 | Starmon | $t, T$ | 2.0 | 2.77 | |
| 6 | Starmon | $t, T$ | 2.5 | 3.26 | |
| 7 | Starmon | $t, T$ | 3.0 | 3.76 | failed after 9 June 19:39 |
| 8 | Starmon | $t, T$ | 3.5 | 4.25 | |
| 9 | Starmon | $t, T$ | 4.0 | 4.74 | |
| 10 | Starmon | $t, T$ | 4.5 | 5.23 | |
| 11 | Starmon | $t, T$ | 5.0 | 5.72 | |
| 12 | Starmon | $t, T$ | 5.5 | 6.23 | |
| 13 | Starmon | $t, T$ | 6.0 | 6.71 | |
| 14 | Starmon | $t, T$ | 6.5 | 7.21 | |
| 15 | Starmon | $t, T$ | 7.0 | 7.70 | broken |
| 16 | Starmon | $t, T$ | 7.5 | 8.19 | |
| 17 | Starmon | $t, T$ | 8.0 | 8.69 | |
| 18 | Starmon | $t, T$ | 8.5 | 9.18 | |
| 19 | Starmon | $t, T$ | 9.0 | 9.68 | |
| 20 | Starmon | $t, T$ | 9.5 | 10.17 | |
| 21 | Starmon, RBR | $t, T, p$ | 10.0 | 10.69 | |
| 22 | Starmon | $t, T$ | 11.0 | 11.66 | |
| 23 | Starmon | $t, T$ | 12.0 | 12.65 | |
| 24 | Starmon | $t, T$ | 13.0 | 13.65 | |
| 25 | Starmon | $t, T$ | 14.0 | 14.64 | |
| 26 | Starmon | $t, T$ | 15.0 | 15.63 | |
| 27 | Starmon | $t, T$ | 16.0 | 16.63 | |
| 28 | Starmon | $t, T$ | 17.0 | 17.63 | |
| 29 | Starmon | $t, T$ | 18.0 | 18.63 | |
| 30 | Starmon | $t, T$ | 19.0 | 19.62 | |
| 31 | Starmon, RBR | $t, T, p$ | 20.0 | 20.63 | |
| 32 | Starmon | $t, T$ | 22.0 | 22.63 | |
| 33 | Starmon | $t, T$ | 24.0 | 24.64 | |
| 34 | Starmon | $t, T$ | 26.0 | 26.65 | |
| 35 | Starmon | $t, T$ | 28.0 | 28.67 | |
| 36 | Starmon, RBR | $t, T, p$ | 30.0 | 30.69 | |
| 37 | Starmon | $t, T$ | 32.0 | 32.71 | |
| 38 | Starmon | $t, T$ | 34.0 | 34.74 | |
| 39 | Starmon | $t, T$ | 36.0 | 36.77 | |
| 40 | Starmon | $t, T$ | 38.0 | 38.81 | |
| 41 | Starmon, RBR | $t, T, p$ | 40.0 | 40.85 | |

**Table 2.** Parameter settings and results of ROMS runs in Series A–E. Bold text indicates those parameters or boundary forcing patterns which are varied within the respective series. The best run of each series is marked by an asterisk and serves as the control run for the successive series.

| Run | $h_c$ [m] | $rx1$ | parent | mixing scheme | $A_{VT}$ [m$^2$ s$^{-1}$] | atmospheric forcing | assimilation | $\Delta T$ [°C] | $\overline{\Delta T}$ [°C] | $\Delta D$ [m] |
|---|---|---|---|---|---|---|---|---|---|---|
| Series A | | | | | | | | | | |
| A1 | 10 | 21 | **MFS** | GLS generic | $1 \times 10^{-6}$ | COSMO-ME | **no** | 0.30 | 1.16 | 3.47 |
| A2 | 10 | 21 | **MERCATOR** | GLS generic | $1 \times 10^{-6}$ | COSMO-ME | **no** | 0.53 | 1.12 | 2.97 |
| A3* | 10 | 21 | **MERCATOR** | GLS generic | $1 \times 10^{-6}$ | COSMO-ME | **yes** | 0.51 | 0.90 | 2.62 |
| Series B | | | | | | | | | | |
| B1* | **10** | 21 | MERCATOR | GLS generic | $1 \times 10^{-6}$ | COSMO-ME | yes | 0.51 | 0.90 | 2.62 |
| B2 | **20** | 27 | MERCATOR | GLS generic | $1 \times 10^{-6}$ | COSMO-ME | yes | 0.49 | 0.89 | 2.67 |
| B3 | **50** | 23 | MERCATOR | GLS generic | $1 \times 10^{-6}$ | COSMO-ME | yes | 0.49 | 0.91 | 2.70 |
| B4 | **100** | 25 | MERCATOR | GLS generic | $1 \times 10^{-6}$ | COSMO-ME | yes | 0.46 | 0.89 | 2.68 |
| B5 | **200** | 27 | MERCATOR | GLS generic | $1 \times 10^{-6}$ | COSMO-ME | yes | 0.44 | 0.89 | 2.75 |
| Series C | | | | | | | | | | |
| C1 | 10 | 21 | MERCATOR | GLS generic | $1 \times 10^{-6}$ | **COSMO-ME** | yes | 0.51 | 0.90 | 2.62 |
| C2 | 10 | 21 | MERCATOR | GLS generic | $1 \times 10^{-6}$ | **COSMO-IT** | yes | 0.42 | 0.98 | 3.45 |
| C3* | 10 | 21 | MERCATOR | GLS generic | $1 \times 10^{-6}$ | **M1** | yes | 0.80 | 0.70 | 3.28 |
| Series D | | | | | | | | | | |
| D1 | 10 | 21 | MERCATOR | **GLS generic** | $1 \times 10^{-6}$ | M1 | yes | 0.80 | 0.70 | 3.28 |
| D2 | 10 | 21 | MERCATOR | **GLS** k-kl | $1 \times 10^{-6}$ | M1 | yes | 0.50 | 0.61 | 2.86 |
| D3 | 10 | 21 | MERCATOR | **GLS** k-$\epsilon$ | $1 \times 10^{-6}$ | M1 | yes | 0.51 | 0.60 | 2.95 |
| D4* | 10 | 21 | MERCATOR | **GLS** k-$\omega$ | $1 \times 10^{-6}$ | M1 | yes | 0.41 | 0.61 | 2.71 |
| Series E | | | | | | | | | | |
| E1 | 10 | 21 | MERCATOR | GLS k-$\omega$ | $\mathbf{1 \times 10^{-6}}$ | M1 | yes | 0.41 | 0.61 | 2.71 |
| E2 | 10 | 21 | MERCATOR | GLS k-$\omega$ | $\mathbf{5 \times 10^{-6}}$ | M1 | yes | 0.38 | 0.62 | 2.74 |
| E3 | 10 | 21 | MERCATOR | GLS k-$\omega$ | $\mathbf{1 \times 10^{-5}}$ | M1 | yes | 0.35 | 0.59 | 2.60 |
| E4 | 10 | 21 | MERCATOR | GLS k-$\omega$ | $\mathbf{2 \times 10^{-5}}$ | M1 | yes | 0.31 | 0.57 | 2.49 |
| E5 | 10 | 21 | MERCATOR | GLS k-$\omega$ | $\mathbf{3 \times 10^{-5}}$ | M1 | yes | 0.31 | 0.56 | 2.36 |
| E6 | 10 | 21 | MERCATOR | GLS k-$\omega$ | $\mathbf{4 \times 10^{-5}}$ | M1 | yes | 0.35 | 0.57 | 2.25 |
| E7 | 10 | 21 | MERCATOR | GLS k-$\omega$ | $\mathbf{5 \times 10^{-5}}$ | M1 | yes | 0.44 | 0.54 | 2.13 |
| E8 | 10 | 21 | MERCATOR | GLS k-$\omega$ | $\mathbf{6 \times 10^{-5}}$ | M1 | yes | 0.49 | 0.56 | 2.11 |
| E9 | 10 | 21 | MERCATOR | GLS k-$\omega$ | $\mathbf{7 \times 10^{-5}}$ | M1 | yes | 0.55 | 0.58 | 2.05 |
| E10 | 10 | 21 | MERCATOR | GLS k-$\omega$ | $\mathbf{8 \times 10^{-5}}$ | M1 | yes | 0.66 | 0.57 | 2.13 |
| E11 | 10 | 21 | MERCATOR | GLS k-$\omega$ | $\mathbf{9 \times 10^{-5}}$ | M1 | yes | 0.72 | 0.55 | 2.08 |
| E12 | 10 | 21 | MERCATOR | GLS k-$\omega$ | $\mathbf{1 \times 10^{-4}}$ | M1 | yes | 0.80 | 0.57 | 2.15 |
| E13 | 10 | 21 | MERCATOR | GLS k-$\omega$ | $\mathbf{2 \times 10^{-4}}$ | M1 | yes | 1.37 | 0.60 | 2.92 |

**Table 3.** Timing and nominal positions of ScanFish tracks considered in this study (cf. Fig. 5). *ROMS analysis* determines the instant of the model output which was used for comparison. *Only the strictly meridional fraction of A10 was utilised.

| track | type | nominal position | start time | end time | duration | ROMS analysis |
|-------|------|------------------|------------|----------|----------|---------------|
| A01 | zonal | 40°06' N | 21 June 14:03 | 21 June 18:15 | 4:12 | 21 June 18:00 |
| A03 | zonal | 40°00' N | 21 June 19:10 | 22 June 00:38 | 5:28 | 22 June 00:00 |
| A05 | zonal | 39°48' N | 22 June 03:00 | 22 June 08:00 | 5:00 | 22 June 06:00 |
| A07 | zonal | 39°36' N | 22 June 12:57 | 22 June 18:05 | 5:08 | 22 June 18:00 |
| A09 | zonal | 39°24' N | 22 June 20:17 | 23 June 01:16 | 4:59 | 23 June 00:00 |
| A10* | meridional | 07°31' E | 23 June 18:20 | 23 June 22:15 | 3:55 | 24 June 00:0h |

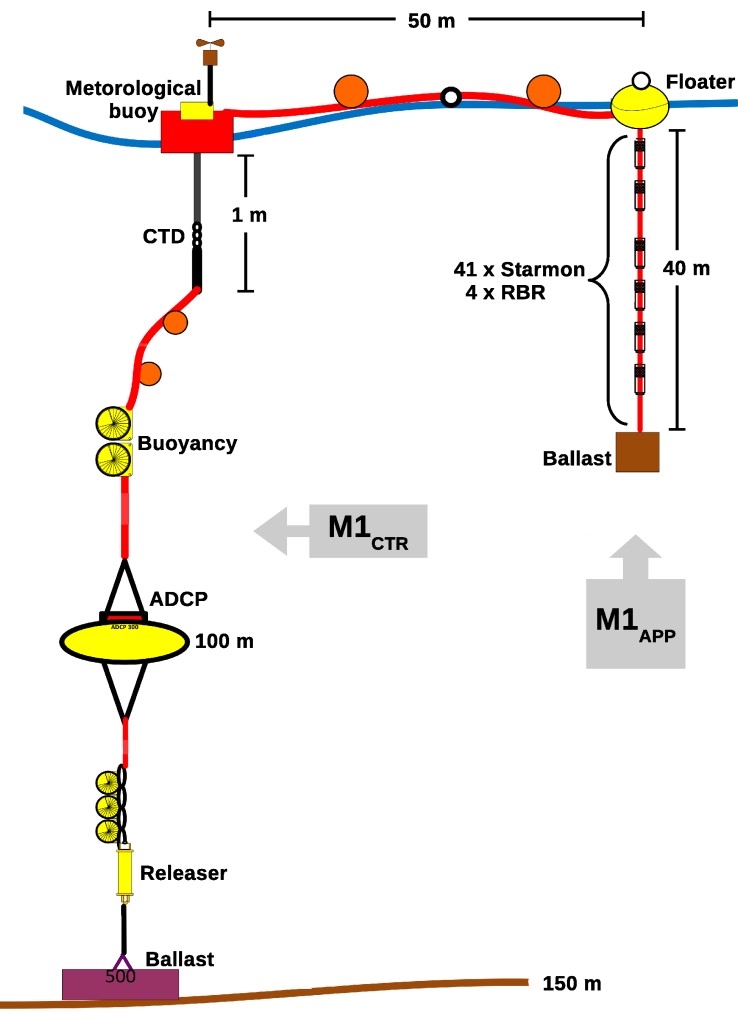

**Figure 1.** Design drawing of mooring M1. For explanations see text.

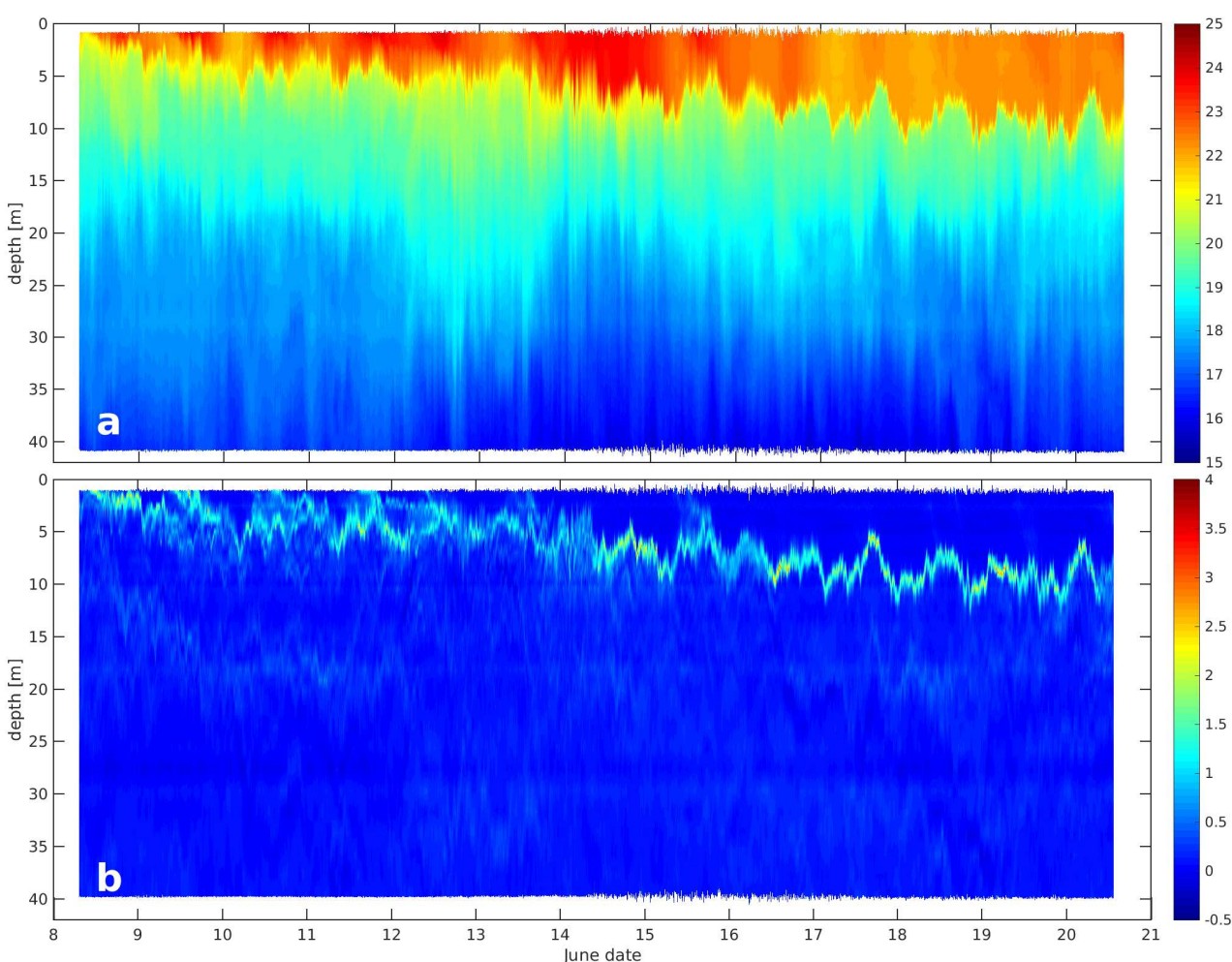

**Figure 2.** (a) Observed temperature [$^\circ$C] at mooring M1 and (b) the vertical temperature gradient [$^\circ$C m$^{-1}$].

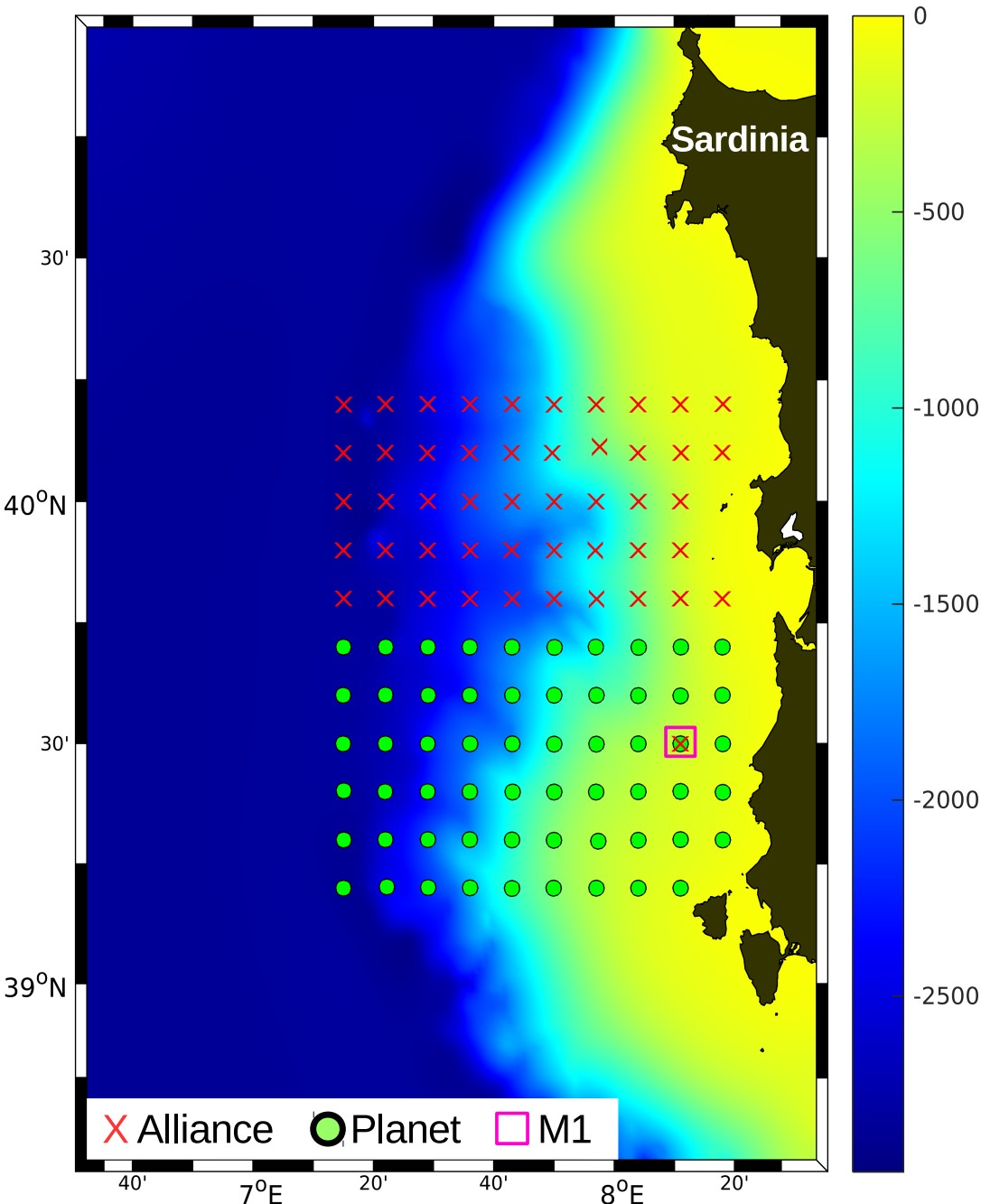

**Figure 3.** CTD casts taken by *Planet* and *Alliance* 7–11 June, and the position of mooring M1. The colourbar indicates the water depth [m].

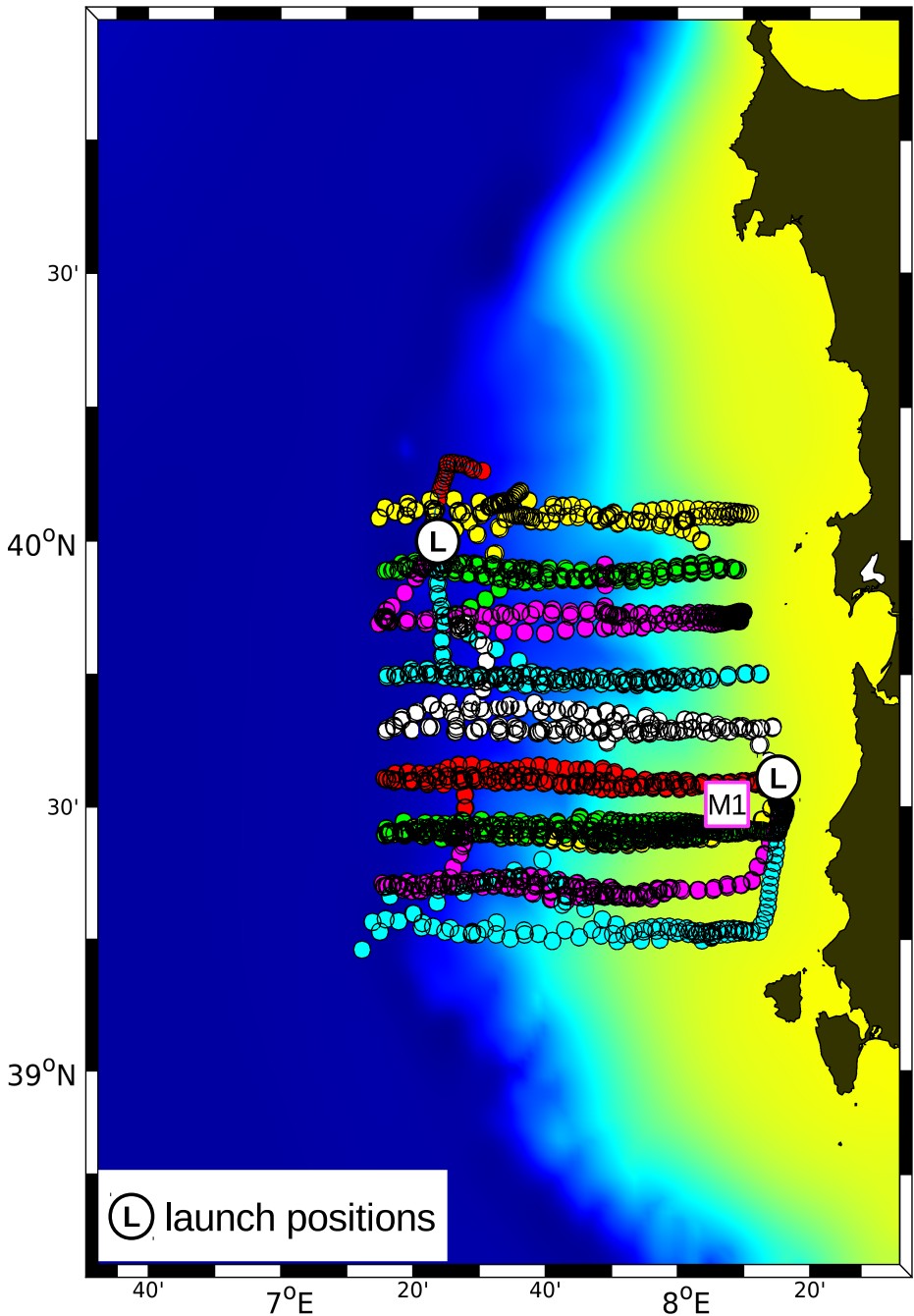

**Figure 4.** Actual glider tracks 8–23 June. The small circles along the tracks show the surfacing positions. Each glider is marked by a different colour. The colour code for the bathymetry is the same as in Fig. 3.

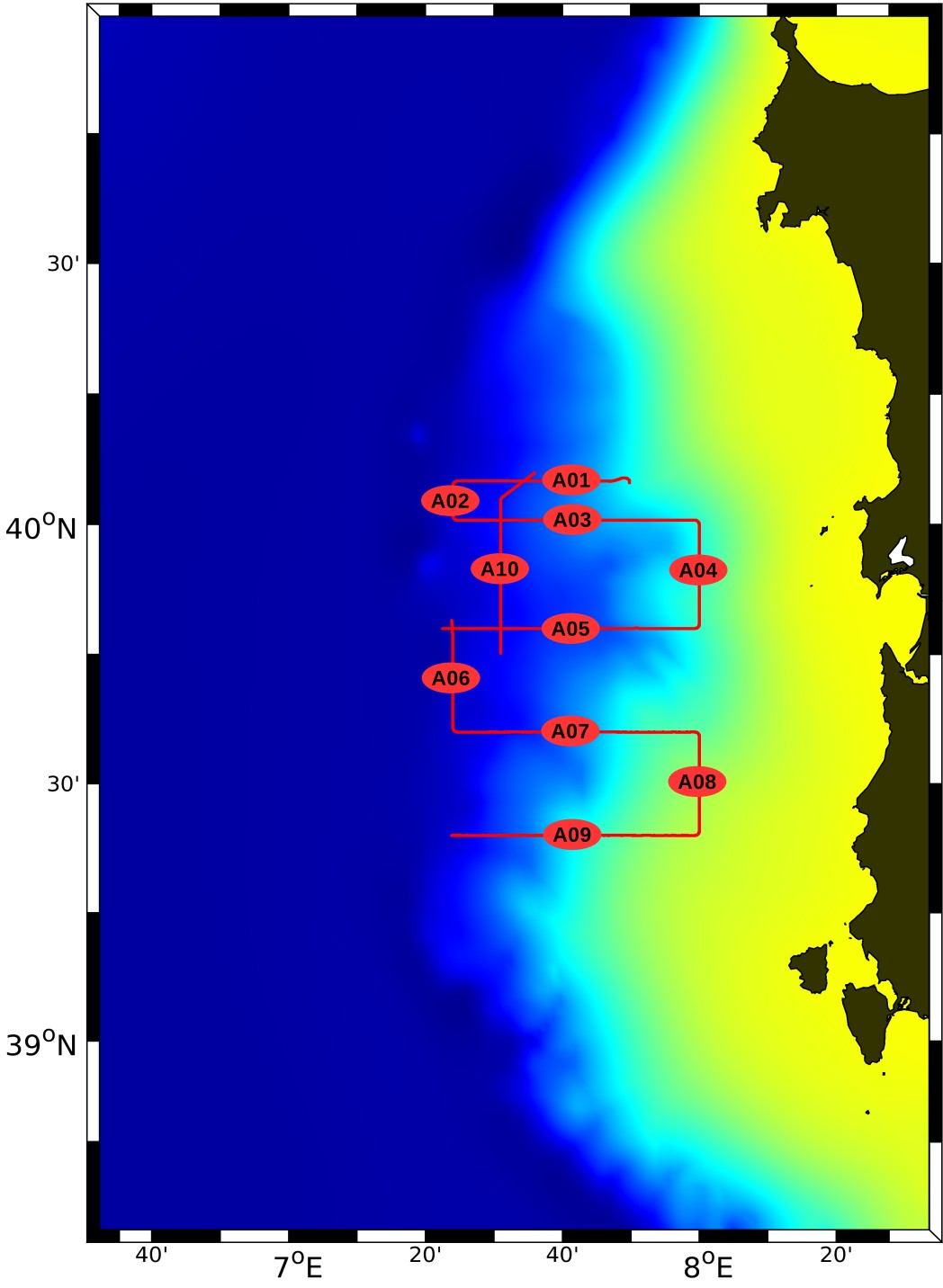

**Figure 5.** ScanFish tracks of *Alliance*, 21–24 June. The colour code for the bathymetry is the same as in Fig. 3

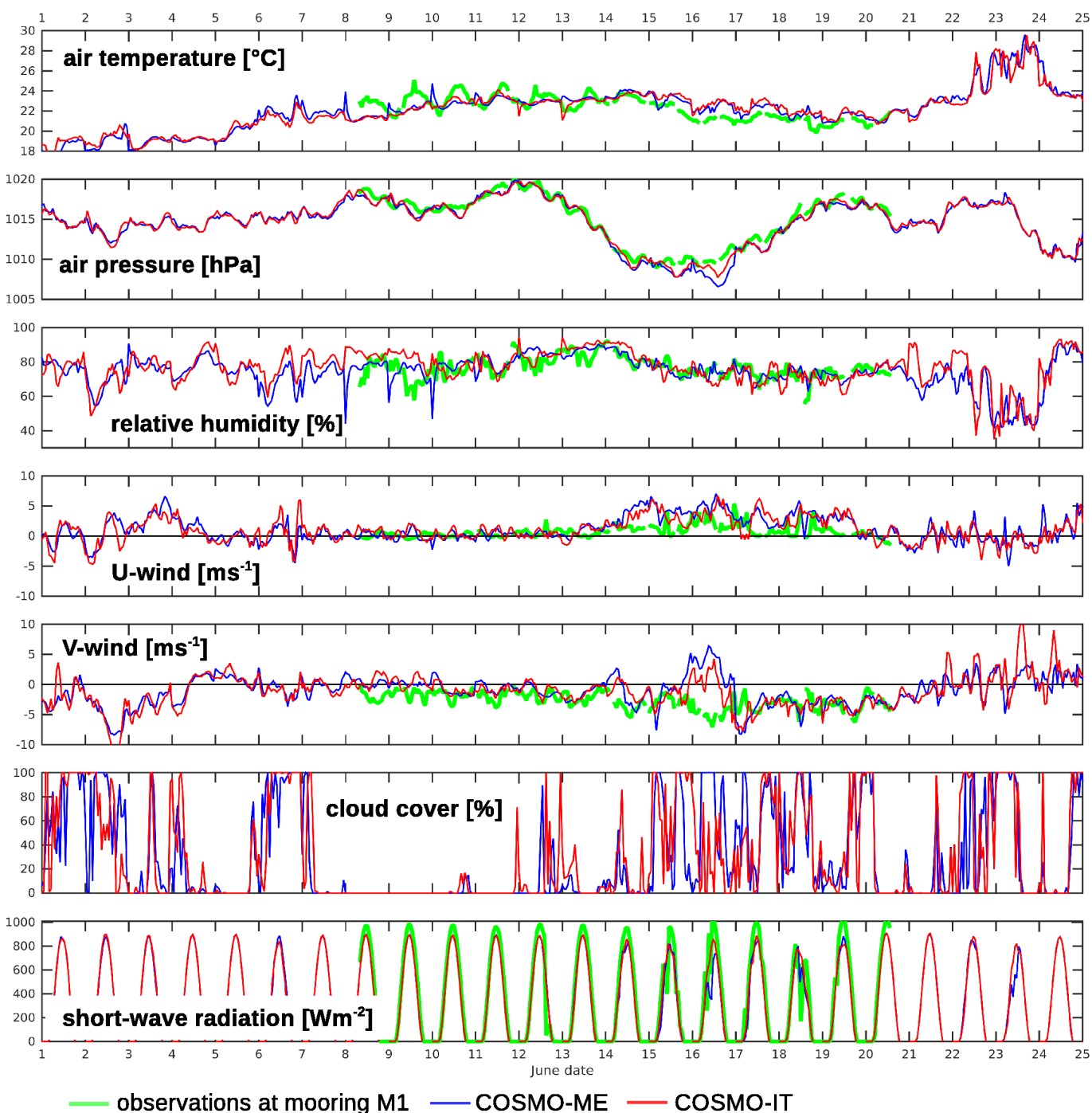

**Figure 6.** Time series of measured and predicted atmospheric parameters at the site of mooring M1 from observations of the meteorological buoy on top of M1, COSMO-ME, and COSMO-IT. *U-wind* and *V-wind* stand for the zonal and meridional wind components, respectively. The cloud cover was not recorded at M1. Precipitation is not shown, because no precipitation was predicted or measured during the entire period.

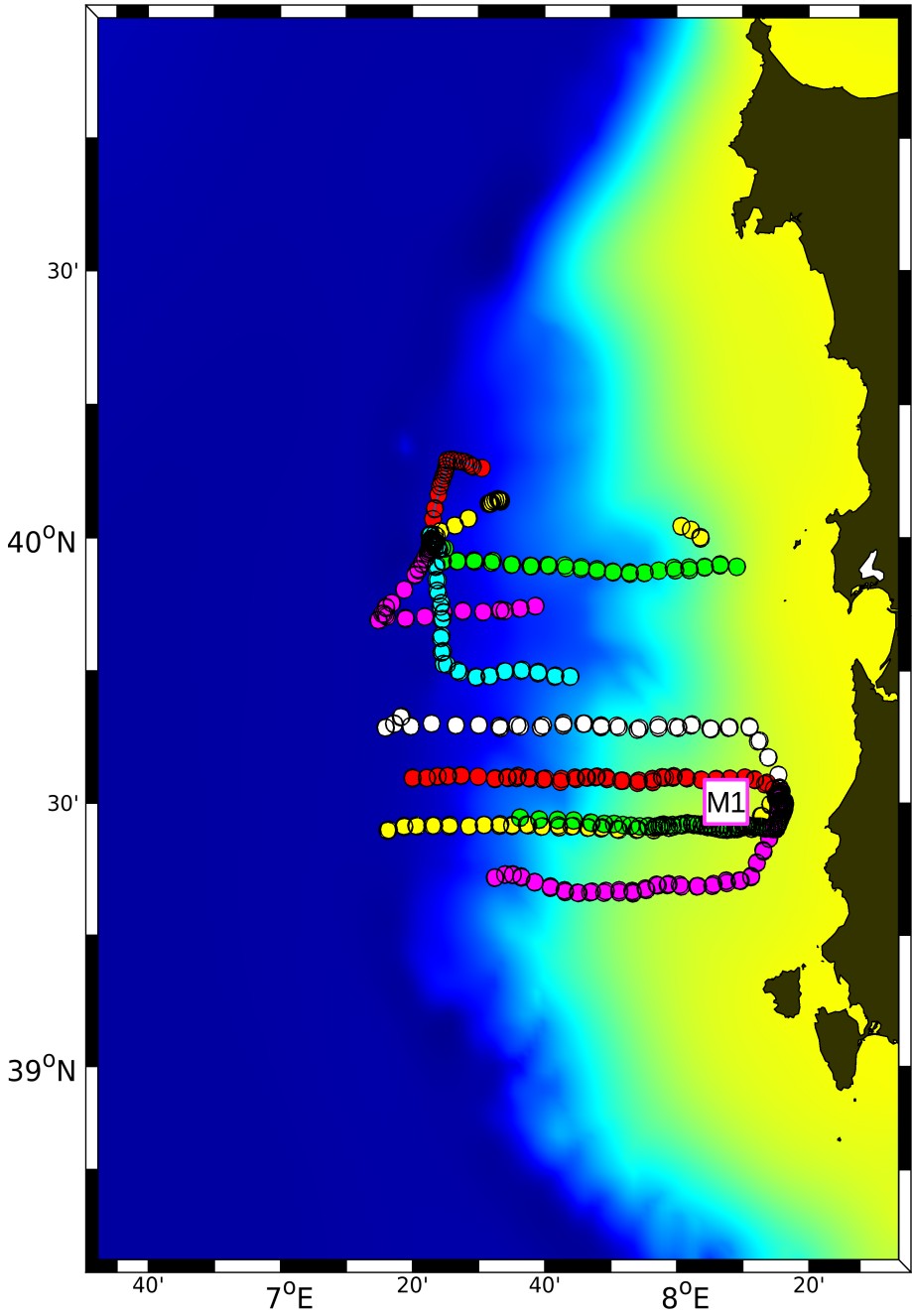

**Figure 7.** Actual surfacing positions of all assimilated gliders 7–11 June. Each glider is marked by a different colour. The colour code for the bathymetry is the same as in Fig. 3.

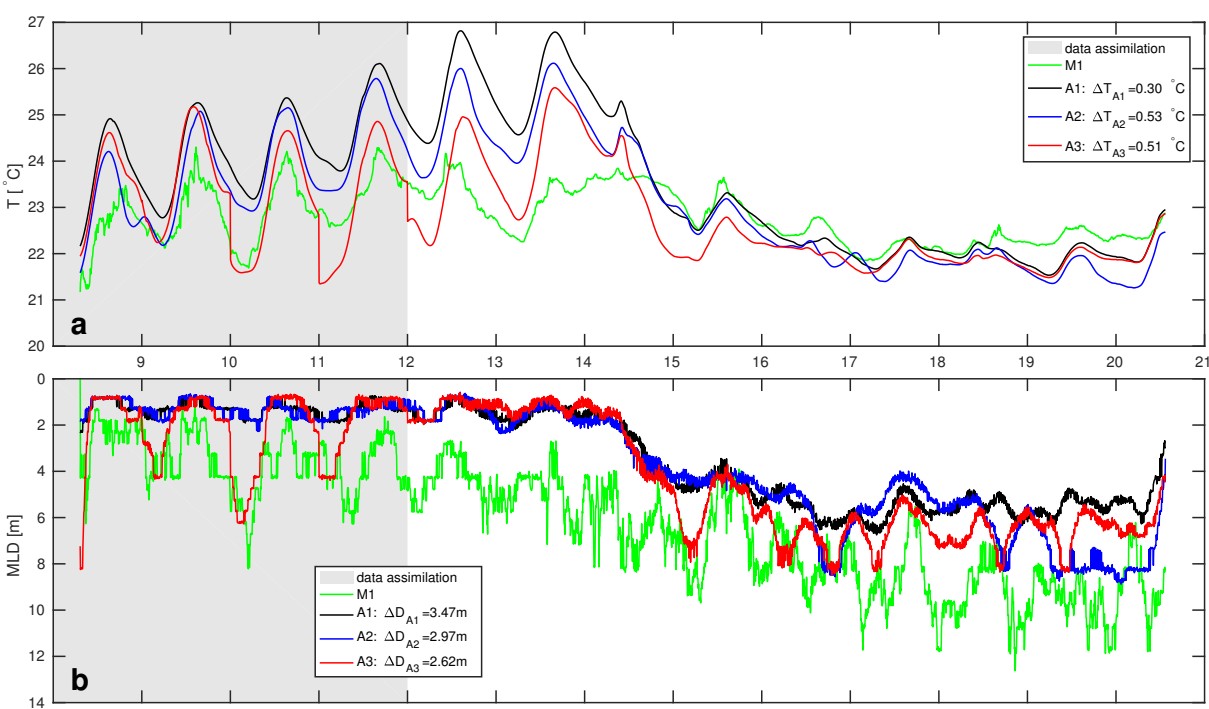

**Figure 8.** ROMS runs A1, A2, A3: time series of (a) near surface temperature at 0.81 m depth and (b) mixed-layer depth (MLD), and the corresponding observations at mooring M1. The numbers on the abscissae indicate June dates. The period where data are assimilated is highlighted by grey shading.

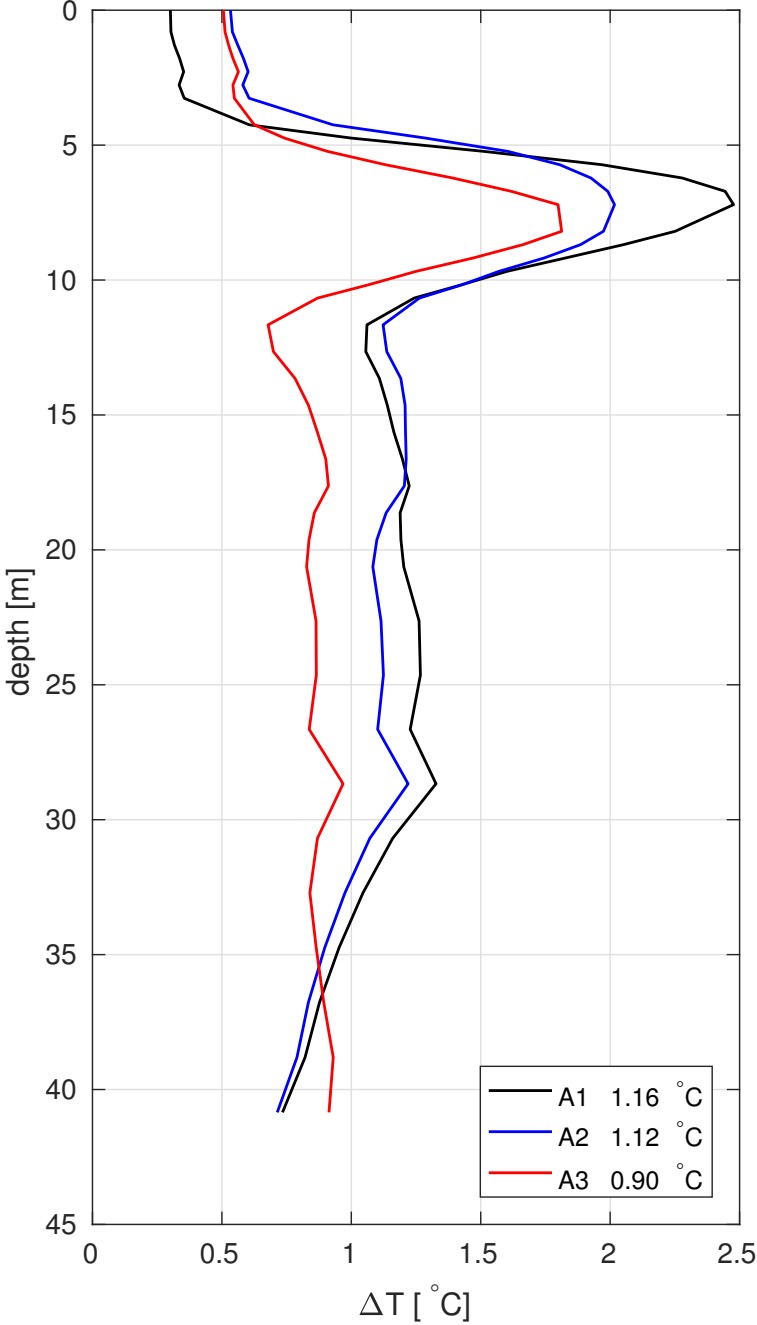

**Figure 9.** ROMS runs A1, A2, A3: *rms* temperature differences $\Delta T$ [°C] between the modelled temperature $T_{ROMS}$ and the observed temperature $T_{obs}$, evaluated at the actual depths of the observations. The vertical mean $\overline{\Delta T}$ is written in the second column of the legend box. $\Delta T$ was computed only for the period after 15 June 00:00

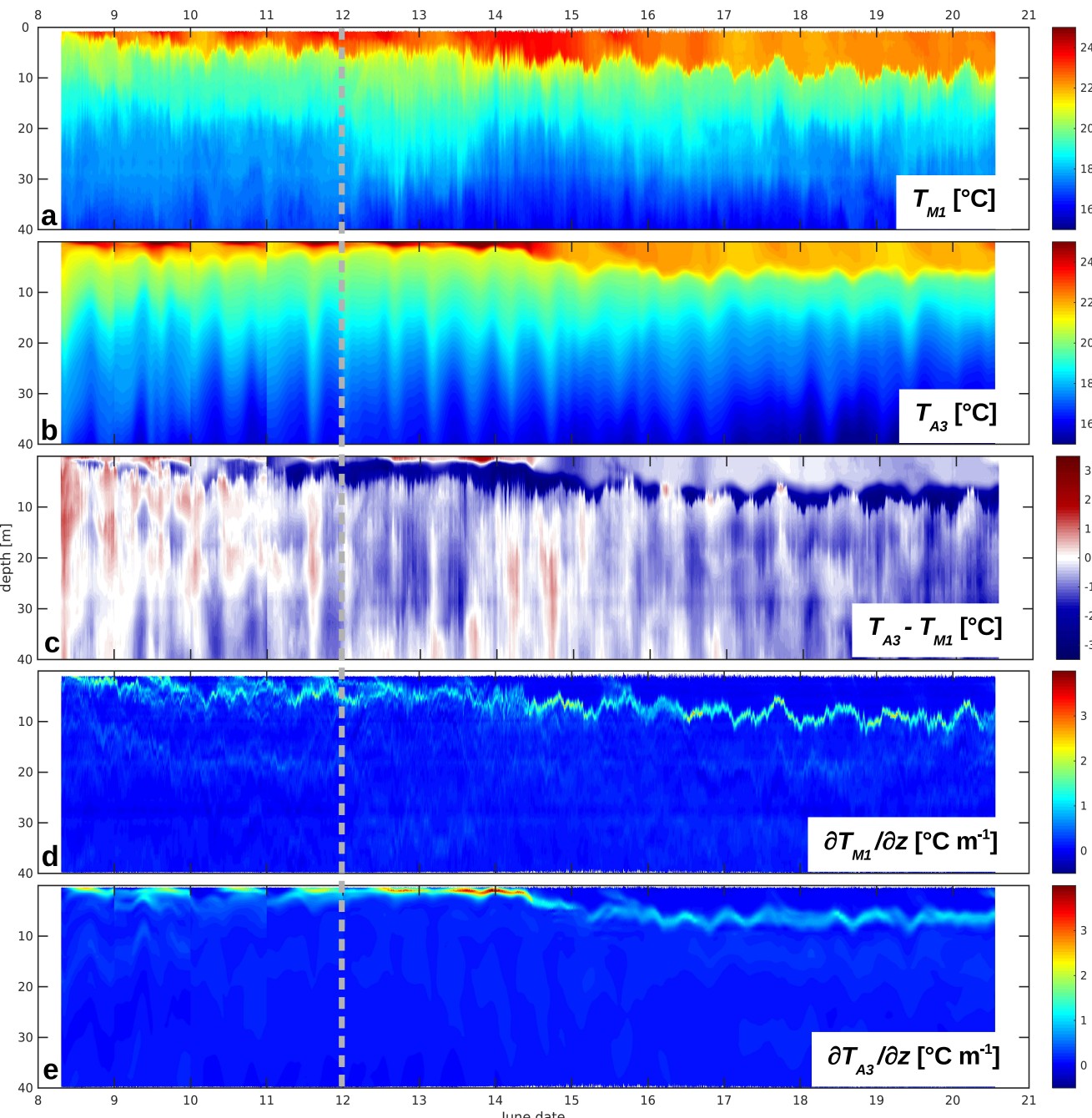

**Figure 10.** (a) Observed temperature at mooring M1, (b) modelled temperature from ROMS run A3, and (c) the difference between the modelled and the observed temperature. The vertical temperature gradient from (d) observations and (e) from A3. The instant of the last data assimilation is indicated by the the grey-dashed vertical line.

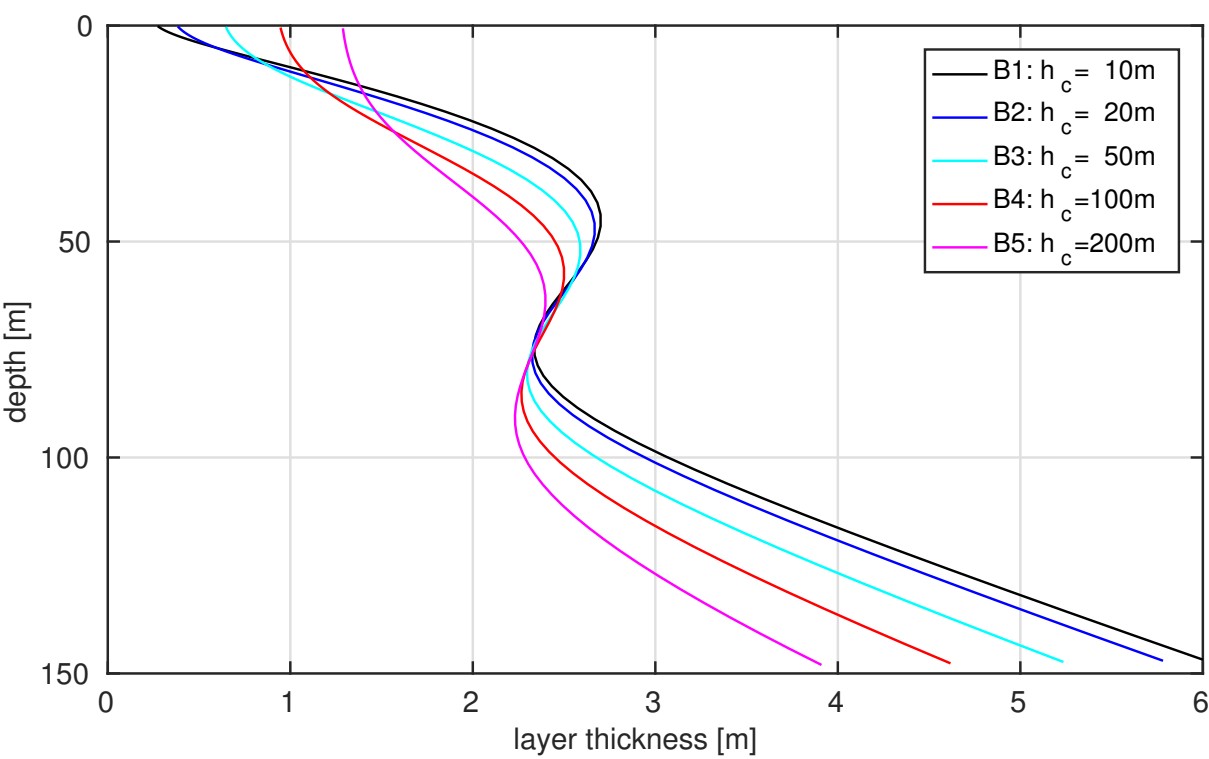

**Figure 11.** The layer thicknesses at the position of mooring M1 for various assumptions of the critical depth $h_c$

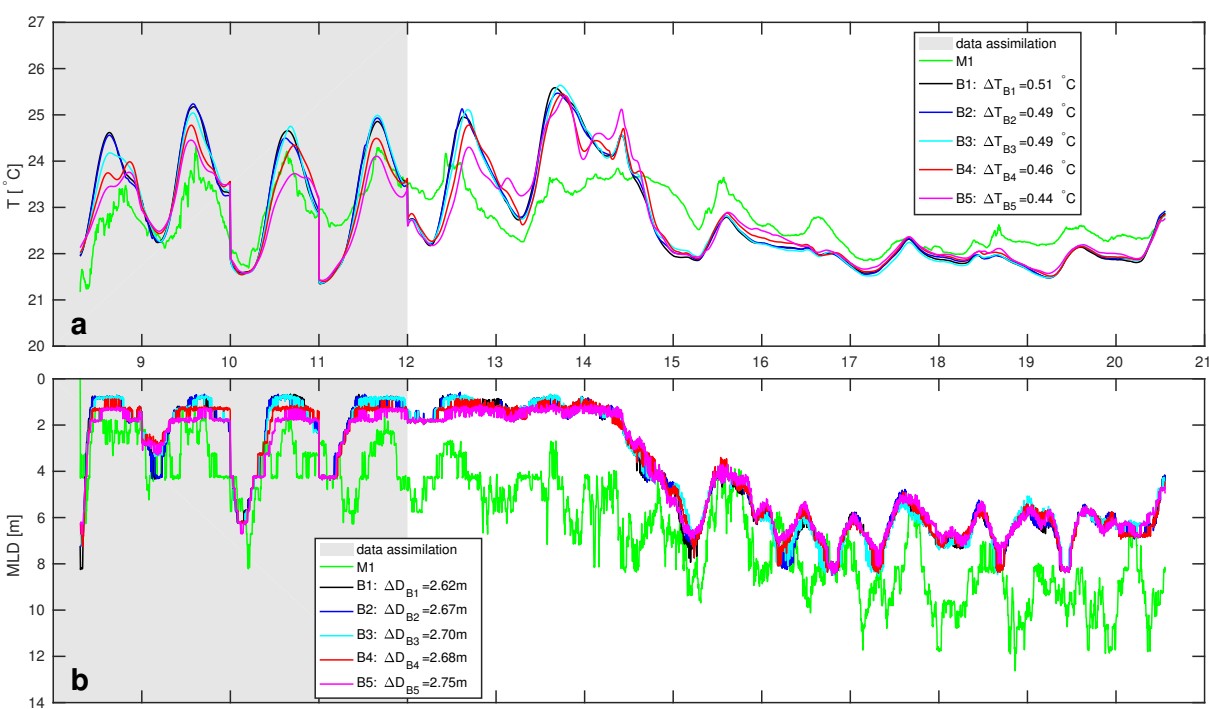

**Figure 12.** ROMS runs B1–B5: time series of (a) near surface temperature at 0.81 m depth and (b) mixed-layer depth (MLD), and the corresponding observations at mooring M1. The numbers on the abscissae indicate June dates. The period where data are assimilated is highlighted by grey shading.

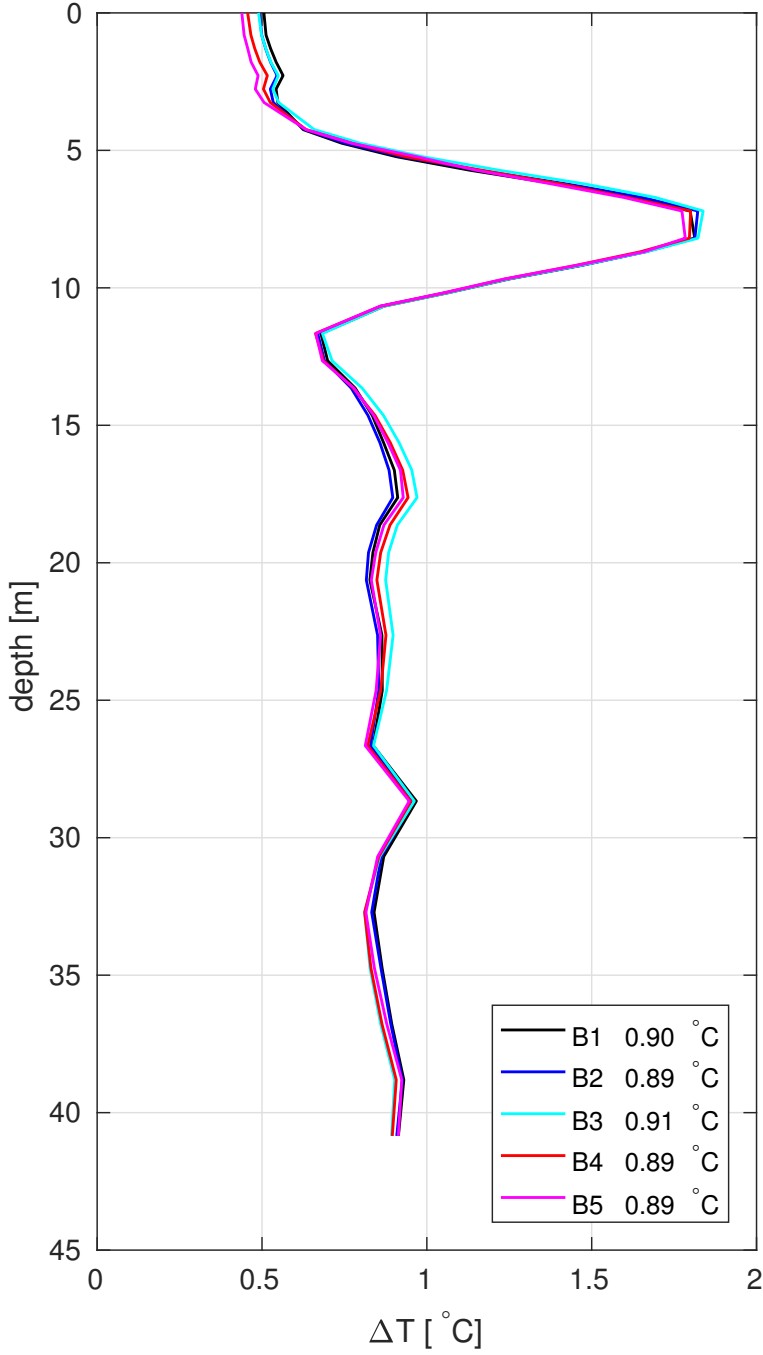

**Figure 13.** ROMS runs B1–B5: *rms* temperature differences $\Delta T$ [°C] between the modelled temperature $T_{ROMS}$ and the observed temperature $T_{obs}$, evaluated at the actual depths of the observations. The vertical mean $\overline{\Delta T}$ is written in the second column of the legend box. $\Delta T$ was computed only for the period after 15 June 00:00.

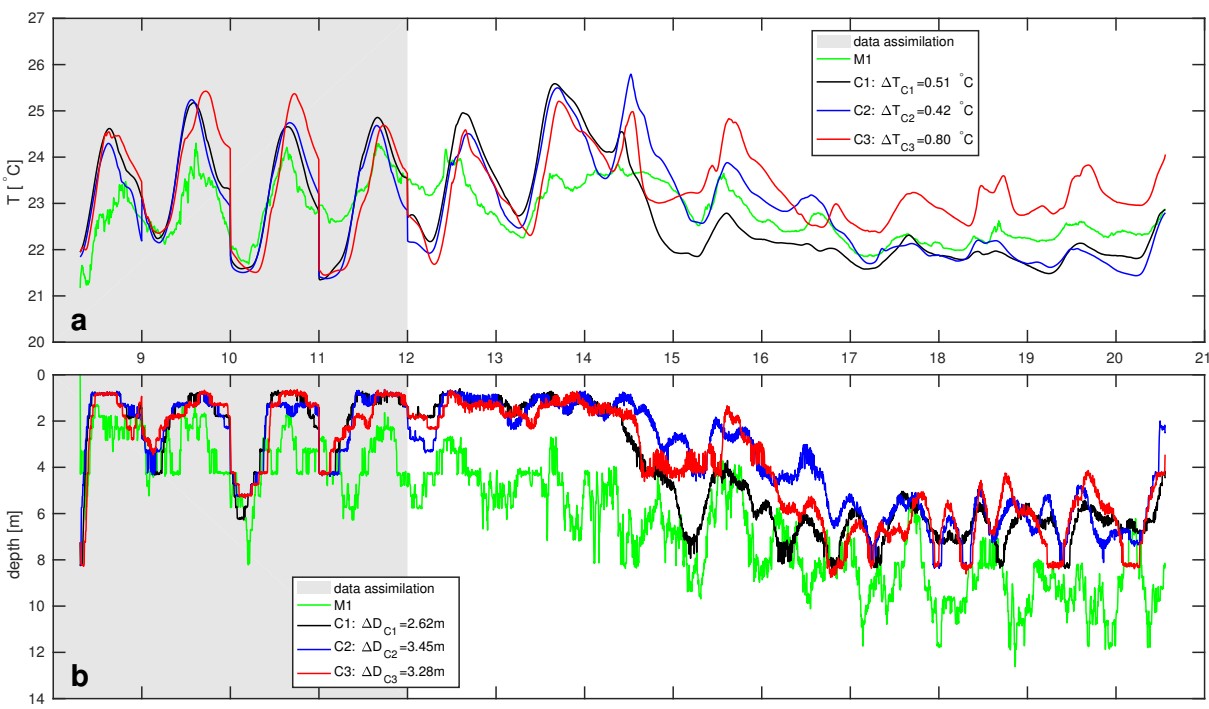

**Figure 14.** ROMS runs C1, C2, C3: time series of (a) near surface temperature at 0.81 m depth and (b) mixed-layer depth (MLD), and the corresponding observations at mooring M1. The numbers on the abscissae indicate June dates. The period where data are assimilated is highlighted by grey shading.

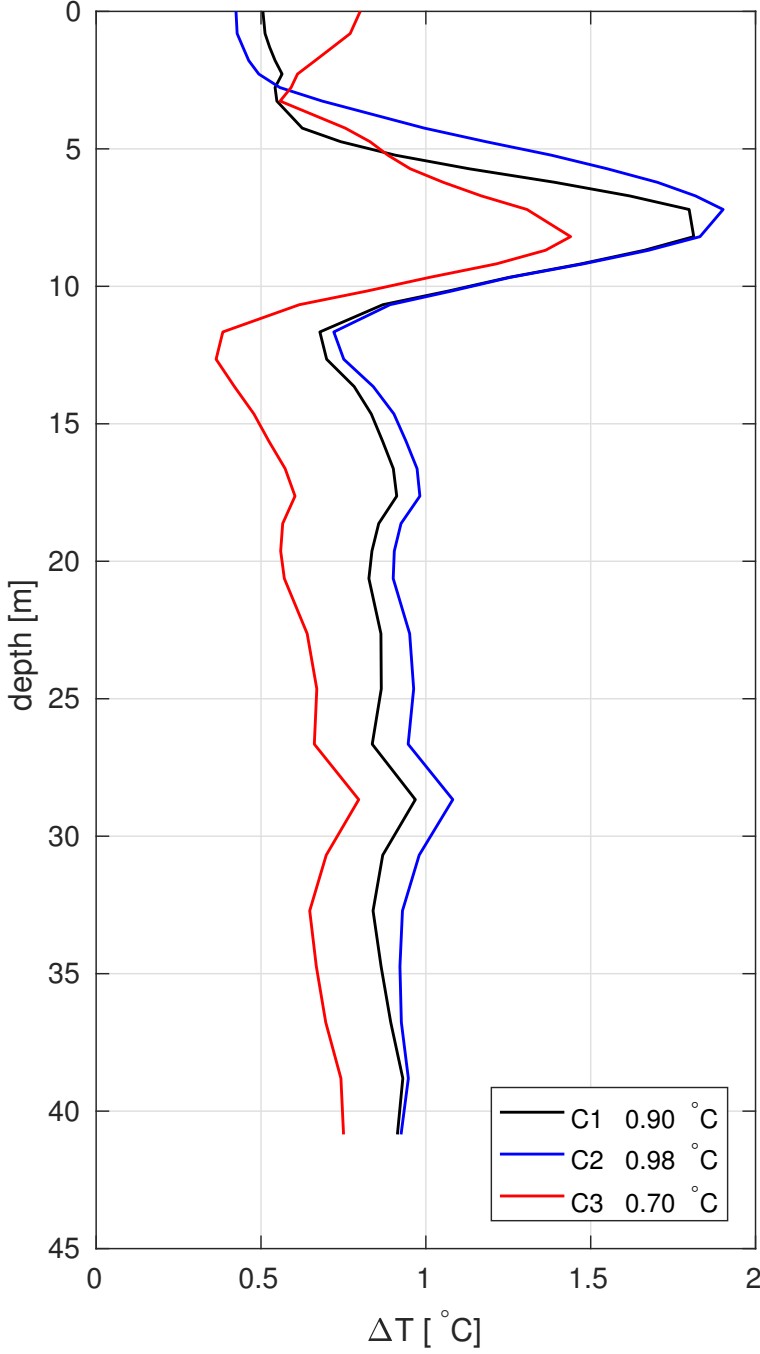

**Figure 15.** ROMS runs C1, C2, C3: *rms* temperature differences $\Delta T$ between the modelled temperature $T_{ROMS}$ and the observed temperature $T_{obs}$, evaluated at the actual depths of the observations. $\Delta T$ was computed only for the period after 15 June 00:00.

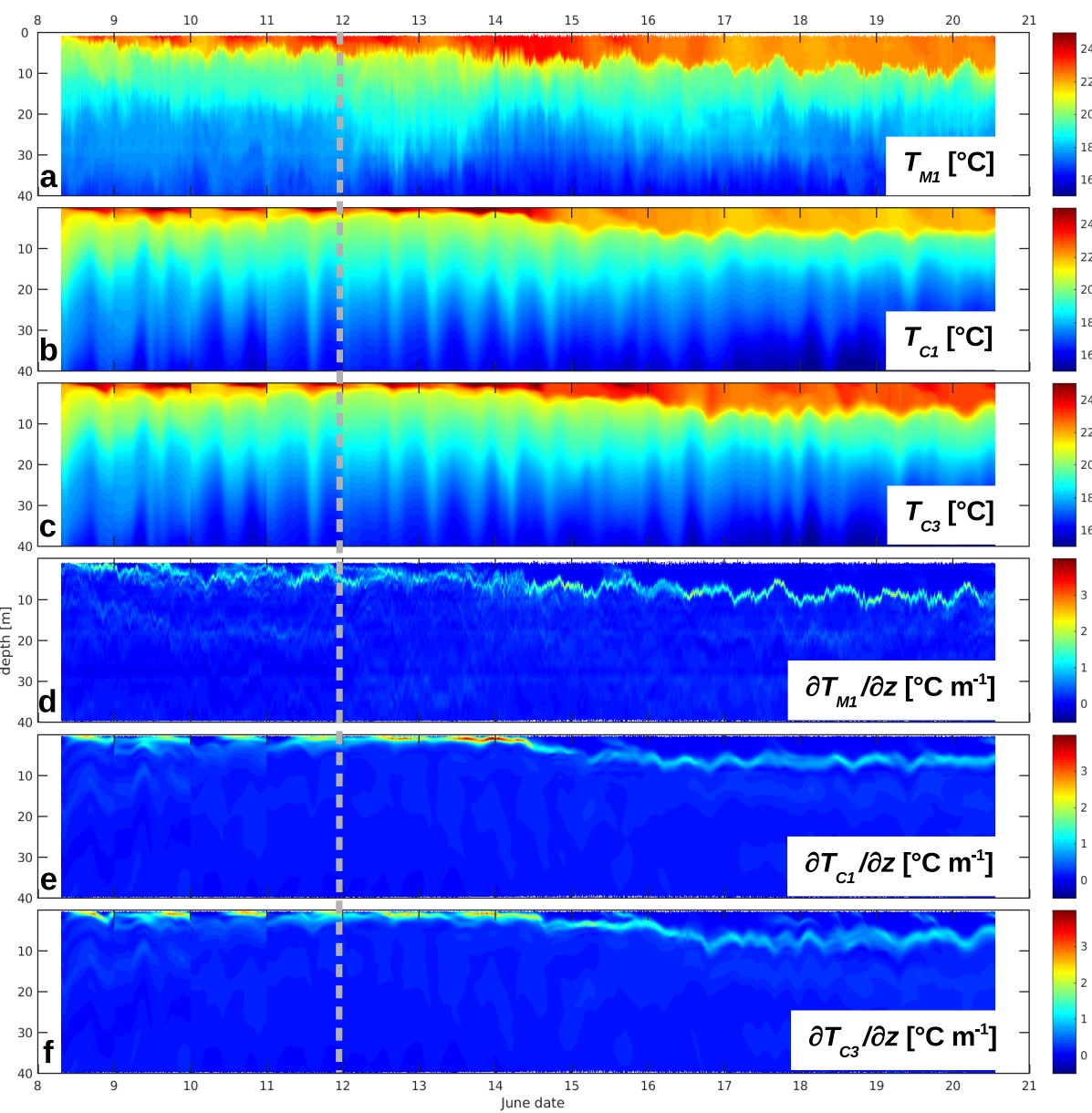

**Figure 16.** (a) Observed temperature at mooring M1,(b) modelled temperature from ROMS run C1 and (c) from C3, (d) the vertical temperature gradient from M1, (e) C1 and (f) C3. The instant of the last data assimilation is indicated by the the grey-dashed vertical line.

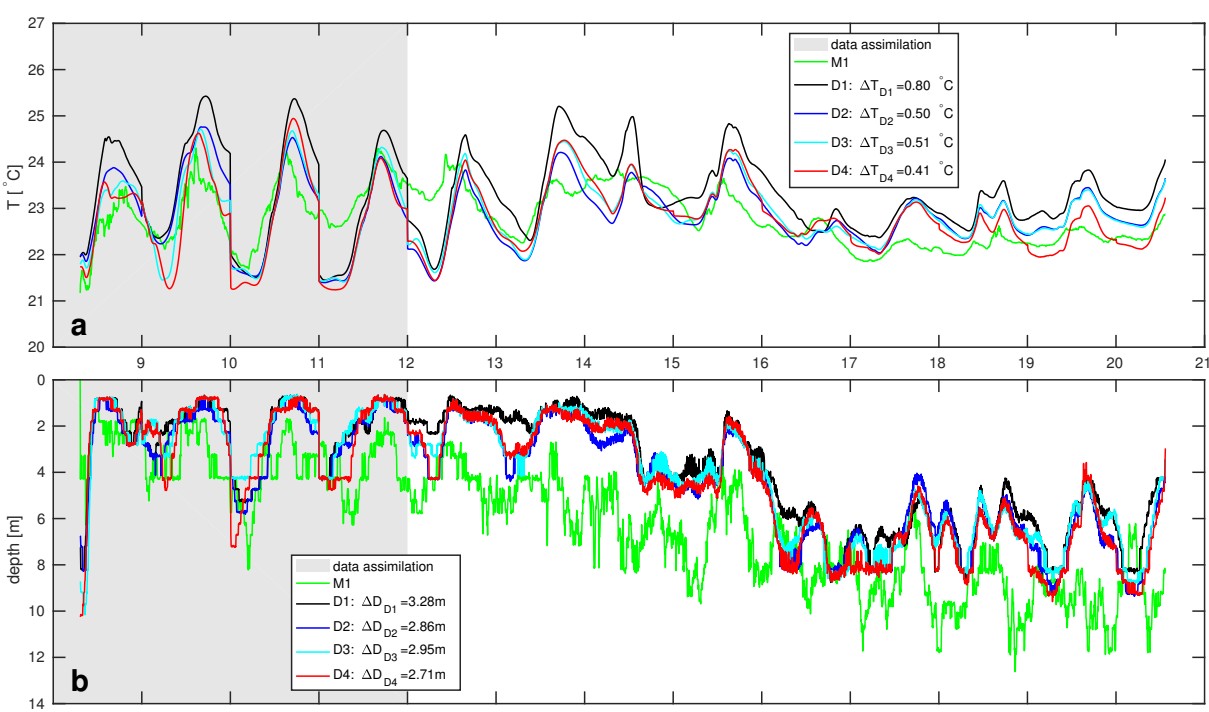

**Figure 17.** ROMS runs D1–D4: time series of (a) near surface temperature at 0.81 m depth and (b) mixed-layer depth (MLD), and the corresponding observations at mooring M1. The numbers on the abscissae indicate June dates. The period where data are assimilated is highlighted by grey shading.

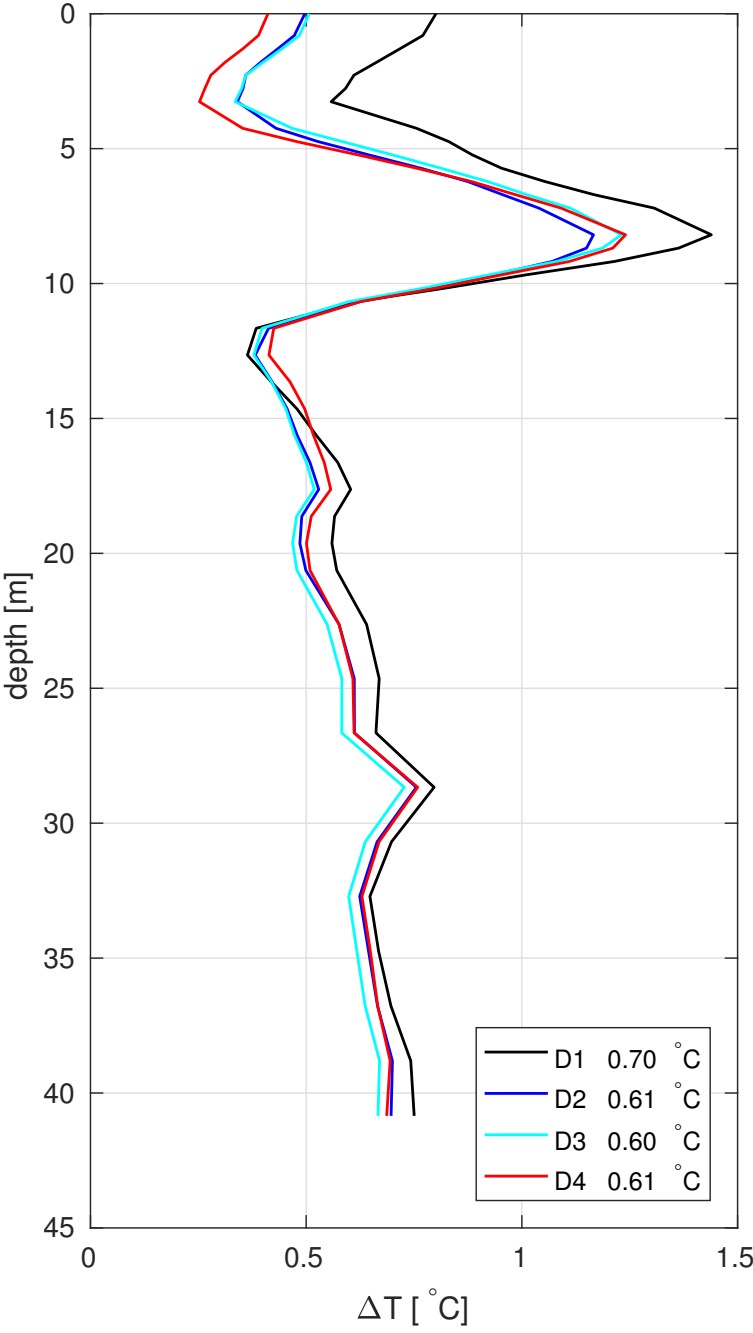

**Figure 18.** ROMS runs D1–D4: *rms* temperature differences $\Delta T$ between the modelled temperature $T_{ROMS}$ and the observed temperature $T_{obs}$, evaluated at the actual depths of the observations. $\Delta T$ was computed only for the period after 15 June 00:00.

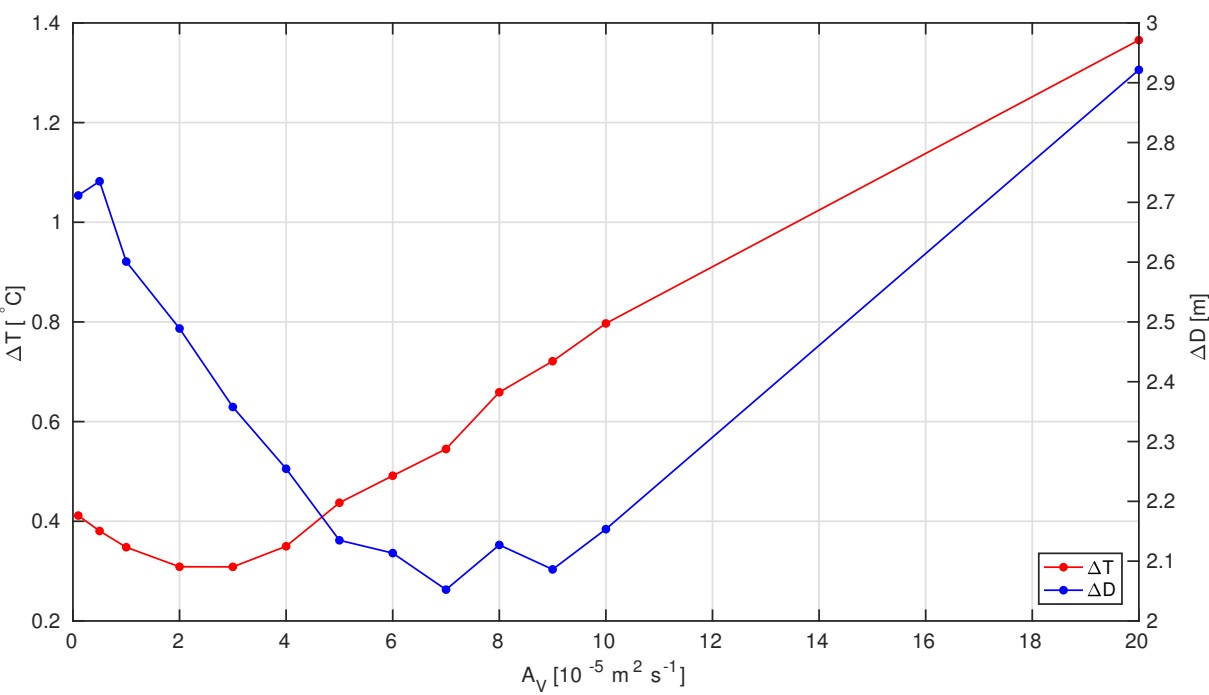

**Figure 19.** Series E: $\Delta T$ and $\Delta D$ for ROMS runs E1 – E13. Both quantities were computed only for the period after 15 June 00:00.

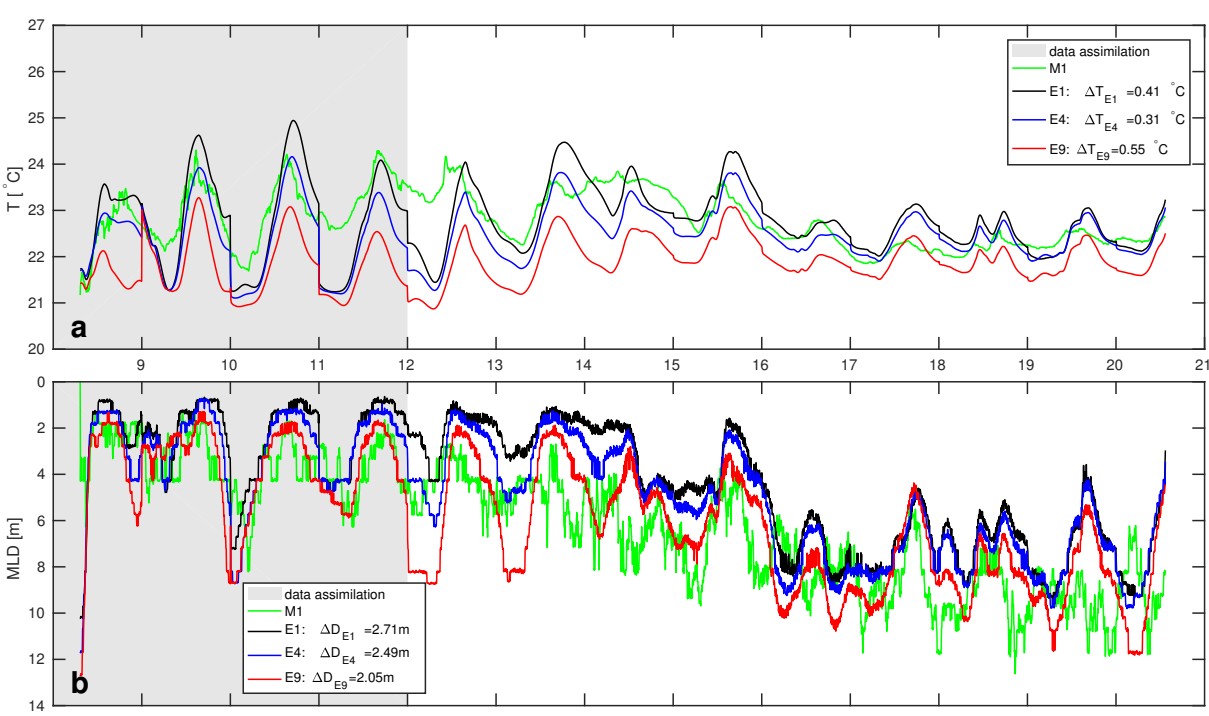

**Figure 20.** Time series of (a) near surface temperature at 0.81 m depth from E1, E4 and (b) mixed-layer depth (MLD) from E1, E9, and the corresponding observations at mooring M1. The numbers on the abscissae indicate June dates. The period time where data are assimilated is highlighted by grey shading.

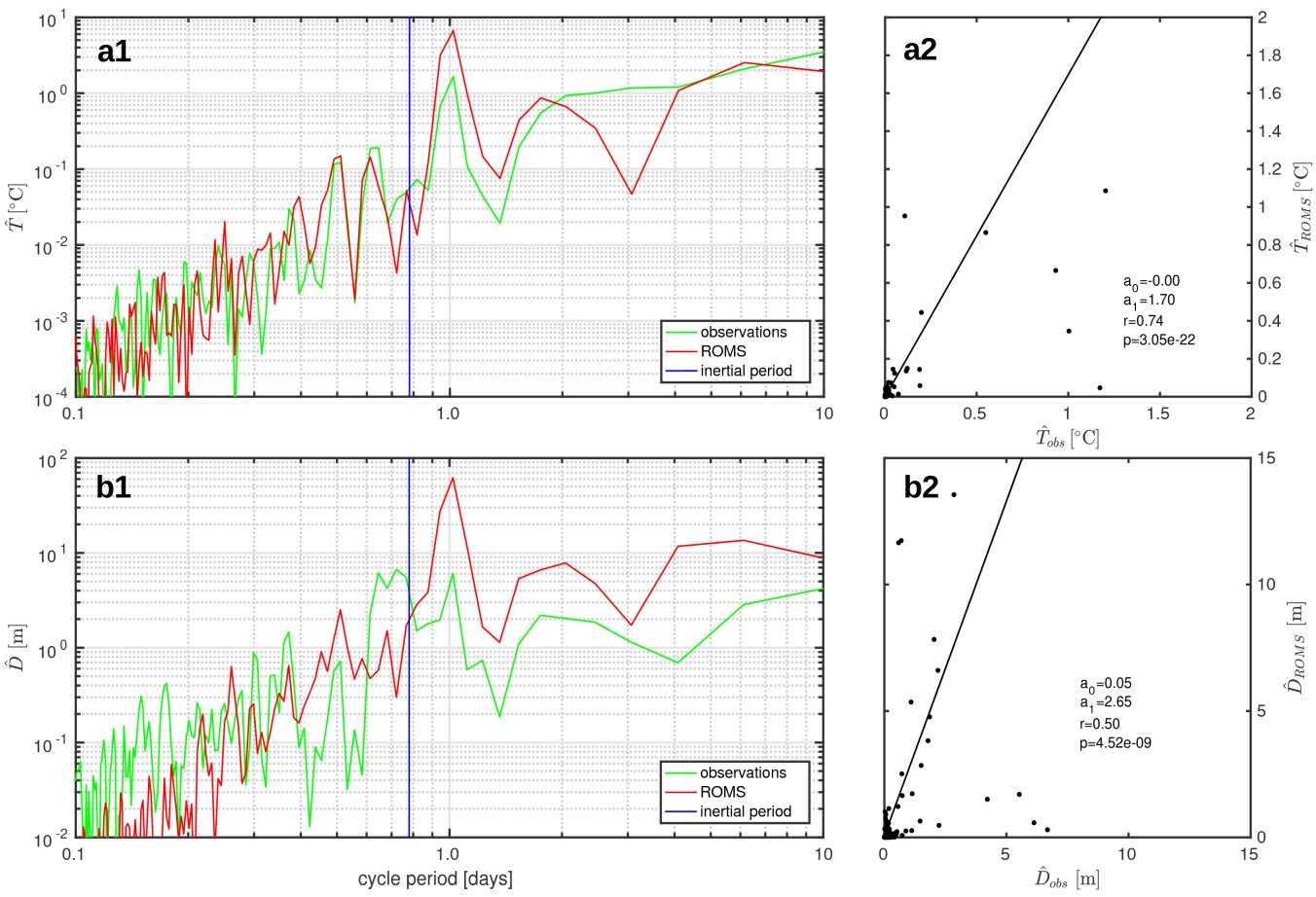

**Figure 21.** Spectra and correlation parameters of the modelled and observed amplitudes (a) $\hat{T}$ of the near-surface temperature at 0.81 m depth from run E4 and (b) $\hat{D}$ of the mixed-layer depth from run E9. The spectra were evaluated for the entire time series 8–20 June where observations were available

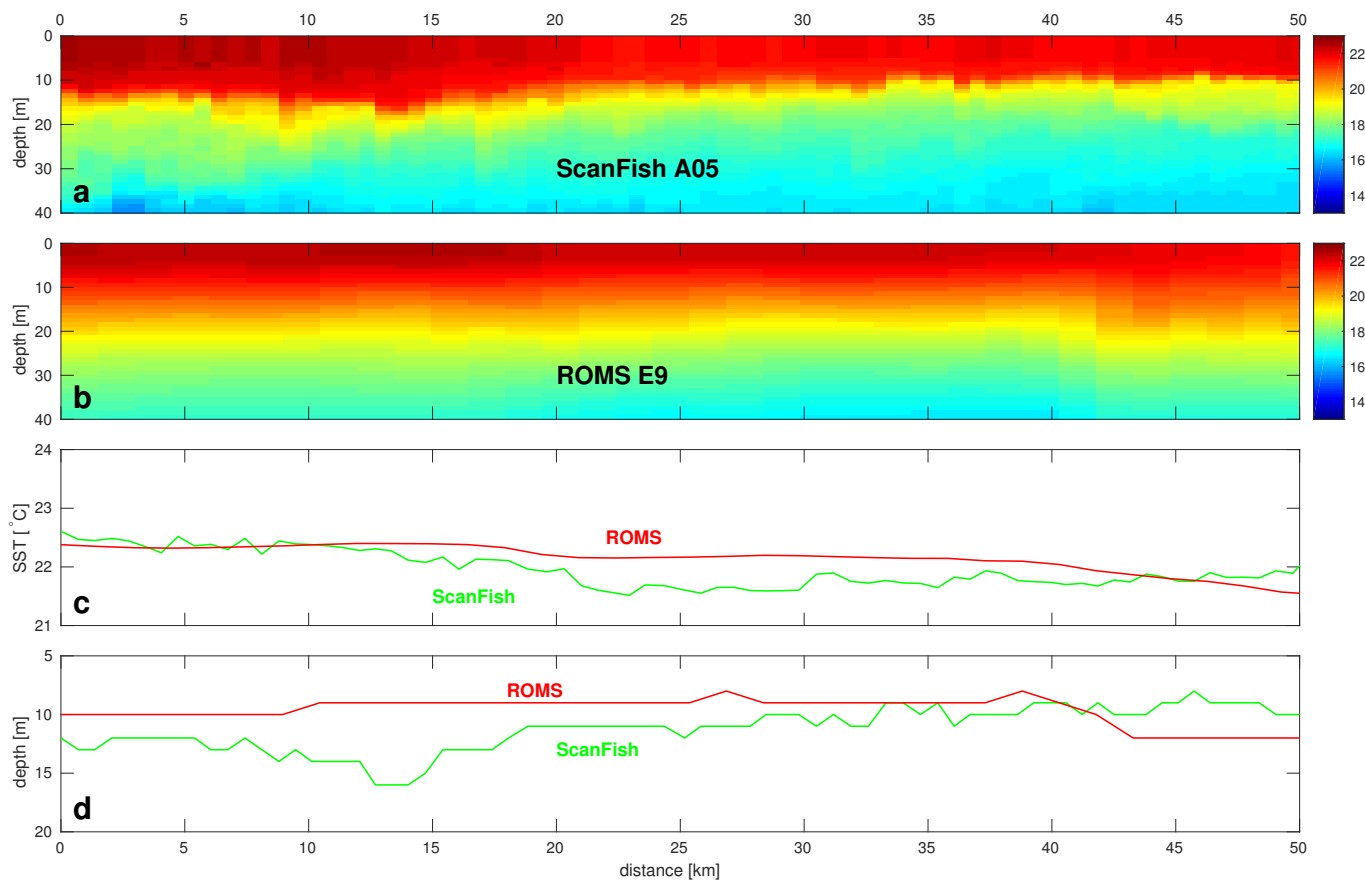

**Figure 22.** Sections along the A05 ScanFish track (cf. Fig. 5) at 40°48'N: (a) temperature recorded by the ScanFish, (b) temperature predicted by ROMS, (c) sea surface temperature SST, and (d) mixed-layer depth evaluated from ScanFish measurements and ROMS. No interpolation was used for the contour plots.