# Peer review of "Validation of an ocean shelf model for the prediction of mixed-layer properties west of Sardinia (Mediterranean Sea)"

_Ocean Science, 2016_

## Referee Comment (RC1) · Anonymous Referee #1 · 7 Nov 2016

**GENERAL COMMENTS**

The manuscript titled "Observed and Modelled Mixed-Layer Properties on the Continental Shelf of Sardinia (Mediterranean Sea)" described the validation of a numerical circulation model ROMS implemented in an area in front of the western coast of Sardinia. The aim is to explore the sensitivity of the forecast skill of local mixed-layer properties to the initial conditions, boundary conditions, and vertical mixing parameterisations. To do this, the author uses the data acquired during the REP14-MED oceanographic survey, realized in June 2015 and supported by NATO, to be assimilated and for model validation. The paper, really technical on ROMS and its limits/potentiality in reproducing local mesoscale and sub-mesoscale variability, is well written and accurately describes all processes that lead to the evaluation of such a model implemented in the area. Can be acceptable for publication after minor revisions.

[Figure]

SPECIFIC COMMENTS

The title " Observed and Modelled Mixed-Layer Properties on the Continental Shelf of Sardinia (Mediterranean Sea)" does not take the reader to the right aim of the paper as no description of properties of the mixed-layer over the Sardinia's shelf is present. I would suggest another title like "Validation of an ocean shelf model for the prediction of mixed-layer properties west of Sardinia (Mediterranean Sea)" or something similar.

The author should verify the use of the units for temperature in the text and figures/captions.

TECHNICAL CORRECTIONS

page 3, line 13: unify the two references Onken et al. in one like (Onken et al., 2014, 2016)

page 4, line 18: correct "begin" in "beginning"

page 5, line 10: unify the two references Shchepetkin and McWilliams in one like (Shchepetkin and McWilliams, 2003, 2005)

page 15, line 9: correct "to shallow" in "too shallow"

page 17, line 17: put up the DOI at the end of the reference of Le Galloudec et al.

page 17, line 24: delete the reference Lurton (2010) as it is not in the text

---

## Referee Comment (RC2) · Anonymous Referee #2 · 16 Dec 2016

**General comments**

The paper by Onken is a large sensitivity study for a numerical setup off the Sardinian coasts (Western Mediterranean Sea). Sensitivity is evaluated with respect to initial and boundary conditions, atmospheric forcings and parameterizations for vertical mixing schemes. The paper deserves to be published as the observational effort to validate and assimilate in the model is impressive: a mooring (with an ADCP, a CTD, 4 RBRs and 40 high-resolution temperature recorders), more than one hundred CTD casts taken by two R/V vessels (Alliance and Planet), deployment of 11 gliders and the use of a ScanFish. However, I do have major and minor critical points (see specific comments below) that I believe should be addressed by the author in the review process.

Specific comments: major concerns

a) My first main criticism of the paper is the following. One would expect such a large observational dataset to be used in a state-of-the-art assimilation scheme. As acknowl-edged by the same author at pag6 (L30-31), ROMS includes a 4DVAR state-of-the-art variational assimilation scheme which can be directly used for sensitivity studies via the analysis of the adjoint of the model. In this work the author uses an Objective Analysis method stating that the variational method is computationally expensive. I believe that at least a quantitative measure of this statement should be provided, considered that in the paper I counted at least 24 different runs: if one run is the same in each subset, we have 3 different initial conditions (A1-A3), 5 different vertical grids (B1-B5), 3 surface conditions (C1-C3), 4 vertical schemes (D1-D4), 14 background vertical diffusivity values (E1-E14). The author should really show that running 24 forward runs is less computationally expensive than running a few simulations with a 4DVAR scheme and perform an analysis of the jacobian matrix for the sensitivity assessment. The author should be quantitative and provide values/estimates (core per hours?) for both the 4DVAR and the OA simulations.

b) My second main point is about the choice of the large horizontal eddy diffusivity and viscosity values in the paper. It is my understanding that a horizontal viscosity value of 50 m2/s is used throughout the manuscript while only the diffusivity is reduced from 10 to 1 m2/s in the final section 6 of the paper. All these numbers look large values to me especially because ROMS can be run relying on just implicit viscosity/diffusivity and for such a fine grid resolution (1.5km x 1.5 km): what are the estimated grid Reynolds and Peclet numbers? The author should justify this choice because all simulations may be too sluggish and strongly undermine his results.

c) My third main point is that there is no comparison/discussion on other observational experiments performed in the past aimed at assimilating similar fields. Important paragraphs citing previous works in the literature should be inserted both in the introduction and the discussion/conclusion section.

d) My forth main point is that methodology in general should be better described.
Please refer to specific points 6, 7, 8, 9, 10 and 11 below.

Specific comments: minor concerns

1) Pag3, L1: "...west of Sardinia." Maybe here a reference to Fig2 could be provided?

2) Pag4, L5-19: Section 3 is methodological while this whole paragraph describes more results. I would move it at the top of section 4. This also helps with the big leap in following Fig.10

3) Pag4, L11: ok deltaT = 1 degC = 1 degK but I would stick to Celsius throught the whole paper (see also many points below)

4) Pag4, L21-22: not clear, maybe rephrase as "here only those casts taken during the 7-11 June period were used".

5) Pag4, L24: please calculate and provide an estimate of the Rossby radius from the many CTD casts you have available.

6) End of Pag4: there is no description about the CTDs, how they were calibrated, used probes and sensors, etc. The gliders' description (Slocum?), their sensors, is missing as well.

7) Pag5, Sec3.2: This section should include the total number of numerical experiments and a reference to a Table were all simulations are summarized.

8) Pag5, L22: please provide values for rx0 (h-parameter) after the bathymetry was smoothed. The rx1 parameters for all simulations should also be listed in the table of point 7 above

9) Pag6, L27: a discussion about the differences between the two COSMO products is missing. For example, winds from the two products shown in Fig10 are very different after June 14. I am also surprised that winds in the period June 14-20 are less strong in the finer product than in the coarser one.

OSD
10) Pag7, L1: the description of the OA method is left to these two dated reference. Please provide a better description with formulae

11) What about the sensitivity of OA to the parameters W and C? I am asking because the isotropic correlation is a strong assumption especially in areas close to the coast where across and along-shelf dynamics may differ the most.

12) Pag7, L23: not clear, maybe rephrase as "The purpose of this section is the investigation of the impacts of:"

13) Pag8, L9: not sure I understand, why lack of salinity measurements? Maybe you do not have them at high-resolution as temperature but you do have 108 CTD casts!

14) Pag8, L9: Once again set deltaT to degC

15) Pag8, L9: why did you use this method and not the maximum vertical gradient as for example in Fig10?

16) Pag8, L18: a Table summaring all experiments is definitely needed (see point 7 above)

17) Pag8, L25: degC not K

18) Pag9, L26-29: not clear and not really able to grasp what the author is trying to say here. Could you rephrase and expand?

19) Pag11, L10 and L14: "much better" not so evident to me. Please provide a quantification for this statement. "Qualitative" criteria as at L14 are not acceptable

20) Pag12, L7: I read the paper more times but I am missing why the 0.81m depth was chosen wrt other depths

21) Pag12, L22: please define the hatted variables via formulae

22) pag12: L29 and L32: not sure to follow the argument here, isn't the 18.7h peak the inertial period?

OSD
23) Pag13, L24: not clear is this the first or the second vertical gridpoint where scalar (rho-grid) are defined?

24) Pag13, L29: this statement is really important as I am concerned that your simulations are too viscous to be realistic (see main point b above).

25) Pag14, L4: "...almost perfectly". I believe this is an overstatement. What about simply say "reproduced well the observed one"?

Figures

26) The meridional and zonal extensions of Figs 2, 3 and 4 should be exactly those of Fig5 to better orient the reader

27) Fig8a, 12a, 14a, 17a and 20a: change label and express deltaT in degC

- 28) Fig9, 13, 15 and 18: x-axis label, express deltaT in degC
- 29) Fig19: y-axis label, express deltaT in degC

30) Fig20: panels a1 and a2, change degK into degC.

31) Fig21: please indicate inertial frequency on both a1 and b1 panels

**Technical corrections**

32) Pag4, L2: turnS out

---

## Author Comment (AC1) · 19 Dec 2016

letter

**(1) Change of title**
The Reviewer may be right although a description is present in the manuscript by means of the M1 time series. On the other hand, I have no problem to change the title as suggested by the Reviewer.

**(2) Use of units for temperature ...**
Does the Reviewer mean the mixture of $°C$ and *K(elvin)?* If so, I used $°C$ for absolute temperatures and $K$ for temperature differences. See
https://en.wikipedia.org/wiki/KelvinUse_in_conjunction_with_Celsius:

*In science and engineering, degrees Celsius and kelvins are often used simultaneously in the same article, where absolute temperatures are given in degrees Celsius, but temperature intervals are given in kelvins. E.g. "its measured value was 0.01028 $°C$ with an uncertainty of 60 $\mu K$."*

*This practice is permissible because the degree Celsius is a special name for the kelvin for use in expressing relative temperatures, and the magnitude of the degree Celsius is exactly equal to that of the kelvin.[10] Notwithstanding that the official endorsement provided by Resolution 3 of the 13th CGPM states "a temperature interval may also be expressed in degrees Celsius",[4] the practice of simultaneously using both "$°C$" and "K" is widespread throughout the scientific world. The use of SI prefixed forms of the degree Celsius (such as "$\mu°C$" or "microdegrees Celsius") to express a temperature interval has not been widely adopted.*

**(3) Unify the two references Onken et al.**
The manuscript was written with LATEX, using the style file provided by OS. For citations, this style file offers only the commands `citet` and `citep` which do not allow to unify more than one reference of the same author in one reference. However, this can be repaired by writing the references manually without usage of `citet` and `citep`. Will be done in the final version.

**(4) Correct "begin" in "beginning"**
OK

**(5) Unify the two references ...**
see (3) above, same issue

**(6) Correct "to shallow" in "too shallow"**
OK

**(7) Put up the DOI ...**

OK

**(8) Delete the reference Lurton (2010) ...**
OK

---

## Author Comment (AC2) · 22 Dec 2016

Thank you very much for your detailed comments. Because of the Christmas vacation time, I will come back to you after 9 January.

Sincerely, Reiner Onken

---

## Author Comment (AC3) · 10 Jan 2017

**Manuscript os-2016-85:**
**"Observed and Modelled Mixed-Layer Properties on the Continental Shelf of Sardinia"**

Reply #2 to Reviewer #2

**(a) 4DVAR vs. Objective Analysis**

This manuscript is a contribution to the REP14-MED Special Issue. A major intent of this Special Issue is to test, evaluate, and compare the performances and forecast skills of different assimilation methods. Amongst others, there will be two other manuscripts applying 4DVAR and the Ensemble Kalman Filter to the same data set.

The author of the 4DVAR paper (A. Funk) has already performed several model runs both using OA and 4DVAR. The major results of his test were:

- Using 2 outer and 50 inner loops, the CPU time using 4DVAR is at least one order of magnitude larger than the CPU time of using OA.

- If data are assimilated only for a few days (and this is the case in the present manuscript where data are only assimilated 7 – 12 June), 4DVAR is **not** superior to OA.

Moreover, a successful implementation of 4DVAR requires a lot of experience, and I know from many colleagues that they failed. I was told that even an experienced modeler from Rutgers University did not succeed to run 4DVAR using a large glider data set. I also doubt that the tasks for this manuscript can be managed by just a "few simulations with a 4DVAR" scheme: 4DVAR is a *variational* scheme which perturbs the initial conditions, surface boundary conditions, and lateral boundary conditions *randomly* and based on some statistical constraints. In this study, however, these conditions are modified *deterministically* and I do not see how 4DVAR could tackle this issue. At best, a (weak constraint) 4DVAR scheme might be applied to Series E to assess the sensitivity to the background vertical diffusivity.

Action: Reference to A. Funk will be added, perhaps as "personal communication"

**(b) large eddy viscosity and diffusivity**

You are right – the horizontal diffusivity and viscosity values are too high. These values originated from precursor model tests with a coarser resolution, and they were not adjusted when the grid was refined.

Action: All 24 model runs will be repeated but using now 5 $m^2s^{-1}$ for the diffusivity and 1 $m^2s^{-1}$ for the viscosity.

**(c) citation of previous works**

Action: references to previous works will be added

**(d) better description of methodology**
see below 6, 7, 8, 9, 10, 11

**(1) − P3, L1: reference to Fig. 2**
The experimental area is described in detail in in Onken et al. (2016) which is
another paper in the REP14-MED Special Issue.
Action: I will add a reference.

**(2) − P4, L5-19: "Section 3 is methodological ..."**
The intention for L5–19 was not to anticipate any results; these lines were just
written to emphasize the challenge for the model attempts to reproduce the
observed features.
Action: Instead of moving the lines to the top of Section 4, it is suggested to
include another figure in this place which shows the observations.

**(3) − P4, L11: Celsius vs. Kelvin**
I used $°C$ for absolute temperatures and $K$ for temperature differences; this is
common use in the scientific world. See
`https://en.wikipedia.org/wiki/Kelvin#Use_in_conjunction_with_Celsius`:
*In science and engineering, degrees Celsius and kelvins are often used simulta-
neously in the same article, where absolute temperatures are given in degrees
Celsius, but temperature intervals are given in kelvins. E.g. "its measured value
was 0.01028 C with an uncertainty of 60 K."*

*This practice is permissible because the degree Celsius is a special name for
the kelvin for use in expressing relative temperatures, and the magnitude of the
degree Celsius is exactly equal to that of the kelvin.[10] Notwithstanding that
the official endorsement provided by Resolution 3 of the 13th CGPM states "a
temperature interval may also be expressed in degrees Celsius",[4] the practice of
simultaneously using both "C" and "K" is widespread throughout the scientific
world. The use of SI prefixed forms of the degree Celsius (such as "C" or
"microdegrees Celsius") to express a temperature interval has not been widely
adopted.*
Action: As also the other Reviewer raised this issue, I will use $°C$ throughout
in the revised version of the manuscript.

**(4) − P4, L21-22: not clear ... rephrase ...**
Action: Agreed.

**(5) − P4, L24: estimate Rossby radius**
This does not appear to be necessary. The very detailed study of Grilli and
Pinardi (1998, MTP News, Vol. 6, p4) provides calculations of the Rossby
radius for the entire Mediterranean and for all seasons.
Action: None

**(6) − end of P4: no description about CTDs ... gliders**
The desired descriptions are included in Onken et al. (2016) which is another
paper in the REP14-MED Special Issue.

Action: I will add a reference.

**(7) − P5, Sec3.2: include number of experiments, Table**
Action: A Table will be provided in the revised version.

**(8) − P5, L22: provide rx0 and rx1 parameters**
Action: Agreed.

**(9) − P6 L27: discuss COSMO**
Action: Agreed.

**(10) − P7, L1 : describe OA method**
OA is a common method in meteorology and oceanography to interpolate data from irregularly spaced locations to a fixed grid. Meanwhile, this is text book stuff (e.g. Thomson and Emery, Data Analysis Methods in Physical Oceanography, Elsevier, 2014, 680 pp.). Therefore, formulae will no be provided.
Action: Up-to-date references will be added.

**(11) sensitivity of OA to W and C**
In a sensitivity study of another paper in the REP14-MED Special Issue (Onken 2017, in preparation), a window size $W$=48 hours was found to provide the best forecast skill.

You are right, the isotropic correlation is a strong assumption especially close to the coast. The OA package used in this study offers only the possibility to use an isotropic correlation or a non-isotropic correlation, specifying the meridional and zonal correlation scales separately. However, the non-isotropic scales will then be applied to each grid point in the model domain which then will bias the results in areas distant from the coast. According to the observations, predominantly meridonal currents are prevailing only in a 10-km wide stripe along the Sardinian coast while the rest of the 180-km wide model domain is characterized by an eddy field with alternating currents. Here, the usage of a non-isotropic correlation scale would definitely deteriorate the results. Hence, an isotropic scale for the entire domain appears to be the best choice.
Action:

- Reference to Onken (2017) will be added
- Choice of isotropy will be discussed.

**(12) − P7, L23: rephrase**
Action: Agreed.

**(13) − P8, L9: lack of salinity measurements**
The validation data set of mooring M1 does not contain salinity measurements. Therefore, only temperature can be validated.
Action: I will make it clearer

**(14) − P8, L9: Celsius vs. Kelvin**
see above (3)

**(15) – P8, L9: why this method?**
The most common methods for definition of the mixed-layer are based on a temperature or density difference criterion. In the main framework of this study, only the temperature criterion is appropriate because salinity (for density calculation) is not available from the observations at M1. The maximum gradient method in Figs. 10 and 16 is only used in order to demonstrate that the gradients are weaker in the model.
Action:

- Definition of mixed-layer depth: references will be added
- Maximum gradient method: no change **or** remove the corresponding images

**(16) – P8, L18: Table**
see above (7)

**(17) – P8, L25: Table**
see above (3)

**(18) – P9, L26–29: rephrase and expand**
Action: Agreed.

**(19) – P11, L10 and L14: "much better" not so evident**
Action: Agreed.

**(20) – P12, L7: 0.81m depth**
0.81 m is the depth of the uppermost Starmon sensor (cf. Table 1).
Action: I will make it more clear in the revised version.

**(21) – P12, L22: define hatted variables**
The hatted variables are the amplitudes. This is written in line 22.
Action: None

**(22) – P12, L29 and L32: inertial peak**
Yes, the inertial peak is (theoretically) at 18.7 hours but it is not detected in the observations. It is not clear to me why you can't follow the argument.
Action: I will check once more.

**(23) – P13, L24: rho-grid**
The first (i.e. the *last* because counting levels starts at the bottom) vertical grid point of the rho-grid is **below** the sea surface. The sea surface is the last grid point of the $w$-grid.
Action: I will make it clear.

**(24) – P13, L29: ... simulations too viscous**
see above (b).

**(25) – P14, L4: "almost perfectly" ...**

Action: Agreed.

**(26) – Figs. 2, 3, 4: meridional and zonal extensions of figures**
Action: This can easily be done.

**(27) – Fig 8a, 12a, 14a, 17a, 20a: Celsius vs. Kelvin**
see above (3)

**(28) – Fig. 9, 13, 15, 18: Celsius vs. Kelvin**
see above (3)

**(29) – Fig. 19: Celsius vs. Kelvin**
see above (3)

**(30) – Fig. 20: Celsius vs. Kelvin**
see above (3)

**(31) – Fig. 21: indicate inertial frequency**
Action: Agreed.

**(32) – P4, L2: turnS out**
Action: Agreed.

---

## Author Response (AR2)

**Action taken on specific points raised by the Topic Editor and Reviewer #2**

(1) **I would specify that we are talking on "gravity" waves. So "Activity of surface gravity waves" or, even better, just "sea state conditions"**
Action: done (see new ms P2L27).

(2) **I am not sure Sardo-Balearic Sea is a geographical name internationally accepted**
A year ago, I discussed this issue already with Aniello Russo (CMRE). Common names in the Literature are *Algero-Provençal Basin* and *West Sardina(n) Sea* but in our opinion, none of these expressions hits the nail on the head. However, then we found the appropriate denomination *Sardo-Balearic Basin* which is frequently (and consistently!) used by geologists. E.g. see Fig. 5.1 on page 204 in: Michard, A., Saddiqui, O., Chalouan, A., and de Lamotte, F. (Eds): Continental Evolution: The Geology of Morocco, Springer-Verlag, 423 pp., 2008.
Action: none.

(3) **Citations of websites should be done according to Copernicus rules**
Action: I modified the citations according to http://www.ocean-science.net/
Copernicus_Publications_Reference_Types.pdf. I hope I did it right because I did not completely understand those rules. See
   • P6L11
   • P6L15
   • P17L6
   • P17L15: here no "last access"

(4) **CMEMS should be properly acknowledged**
Action: done (P17L14-15).

(5) **Instead of "cloudiness" please use "total cloud cover" and specify whether the shortwave radiation is net or not**
Action: done (P7L8).

(6) **Using forum discussions ... I would erase the statement from "Namely" to "ROMS run" (i.e., last four lines of section 4.2).**
Action: The corresponding statement was removed and the previous sentence was modified, including the reference to Haney (1991). See P11L17

(7) **"that the \*selected\* parameterization …. In addition, you should state which stability function you have choosen.**
Action:
   • inserted *selected* (P9L22)
   • Kantha-Clayson stability function mentioned (P5L25)

(8) **These lines are awkward. I suggest to rephrase as "Here, the observational data from a June 2014 oceanographic survey are used.…"**
Action: done (P3L26-27)

(9) **Adjective, either you say "6-hour intervals" (no plural in hourS) or "intervals of 6 hours"**
Action: This expression and corresponding expressions were corrected
   • P7L11 (*72-hour forecast*)
   • P8L12, P15L5-6 (*6-hour intervals*)
   • P14L17 (*12-hour peak*)

Kantha1994Kantha1994Kantha and Clayson(1994)

[revised manuscript text omitted]
*, and the MFS and MERCATOR data sets were downloaded from the *Copernicus Marine Environment Service* (`http://marine.copernicus.eu`). REP14-MED was sponsored by *HQ Supreme Allied Command Transformation* (Norfolk, VA, USA).

[revised manuscript text omitted]